# Augmented temperature fluctuation aggravates muscular atrophy through the gut microbiota

Ya Liu[1], Yifan Guo[1], Zheyu Liu[1], Xu Feng[1], Rui Zhou[1], Yue He[1], Haiyan Zhou [1], Hui Peng[1] & Yan Huang[1,2] ✉

Large temperature difference is reported to be a risk factor for human health. However, little evidence has reported the effects of temperature fluctuation on sarcopenia, a senile disease characterized by loss of muscle mass and function. Here, we demonstrate that higher diurnal temperature range in humans has a positive correlation with the prevalence of sarcopenia. Fluctuated temperature exposure (10–25 °C) accelerates muscle atrophy and dampens exercise performance in mid-aged male mice. Interestingly, fluctuated temperature alters the microbiota composition with increased levels of *Parabacteroides_distasonis*, *Duncaniella_dubosii* and decreased levels of *Candidatus_Amulumruptor*, *Roseburia*, *Eubacterium*. Transplantation of fluctuated temperature-shaped microbiota replays the adverse effects on muscle function. Mechanically, we find that altered microbiota increases circulating aminoadipic acid, a lysine degradation product. Aminoadipic acid damages mitochondrial function through inhibiting mitophagy in vitro. And *Eubacterium* supplementation alleviates muscle atrophy and dysfunction induced by fluctuated temperature. Our results uncover the detrimental impact of fluctuated temperature on muscle function and provide a new clue for gut-muscle axis.

Climate change has a profound effect on human health[1]. Recently, it has been reported that fluctuated temperature contributes to several adverse health outcomes. Diurnal temperature range (DTR), a meteorological indicator reflecting intraday temperature variability[2], is documented to be an independent risk factor for daily mortality in different regions, including China[3], Japan[4], and England[5], etc. Increased DTR also functions as a potential trigger for childhood asthma[6], multiple sclerosis[7], acute upper respiratory tract infections[8]. In addition, large daily temperature fluctuation promotes blood pressure[9] and arrhythmia[10], leading to the occurrence of cardiovascular diseases[11,12]. However, how DTR affects physiological status remains largely unclear.

Sarcopenia, a progressive skeletal muscle disorder, is characterized by loss of muscle mass and strength[13,14]. Multiple factors contribute to sarcopenia, including mitochondrial dysfunction, reduced motor unit and satellite cell number, altered proteostasis, enhanced inflammation[15,16], etc. Increasing evidence reveals that gut microbiota is involved in the onset of sarcopenia. Colonization of gut microbiota from pigs leads to resemblance of muscle fiber structure to pigs in germ-free mice[17]. Mice transplanted with microbiota from high-functioning older adults have improved muscle strength, compared to those transplanted with fragile ones[18]. During aging, altered microbiota composition and dysfunctional gut mucosal barrier lead to increased bacterial products or microbes themselves in circulation[19,20]. Microbiota- derived metabolites exert different effects on skeletal muscle. For example, increased indoxyl sulfate promotes the expression of atrophy-related genes[21], while short-chain fatty acids (SCFAs) improve mitochondrial activity[22]. However, whether and how fluctuated temperature affect microbiota remain unknown.

[1]Department of Endocrinology, Endocrinology Research Center, Xiangya Hospital of Central South University, Changsha, Hunan, China. [2]National Clinical Research Center for Geriatric Disorders, Xiangya Hospital of Central South University, Changsha, Hunan, China. ✉e-mail: yanhuang1018@csu.edu.cn

Here, we attempt to investigate the potential effect of fluctuated temperature on muscle and the possible involvement of gut microbiota in this process. By analyzing the meteorological and demographic data in Chinese, we found that DTR in different cities was positively correlated with the prevalence of possible sarcopenia. In mouse model, fluctuated temperature, ranging from 10 °C to 25 °C, aggravated muscle atrophy in mid-aged mice. These detrimental effects were mediated by altered gut microbiota, and transplantation of which reproduced the adverse effects on muscle. Mechanistically, we found that altered gut microbiota enhanced circulating aminoadipic acid level through the lysine degradation pathway. Elevated aminoadipic acid deteriorated mitochondrial function of muscle through inhibiting mitophagy in vitro. Supplementing *Eubacterium*, the decreased microbiota in feces of fluctuated temperature-treated mice, can partly prevent the deleterious outcome induced by fluctuated temperature. These findings suggest that fluctuated temperature accelerates the onset of sarcopenia in middle-aged individuals, in which gut microbiota plays a critical role.

## Results

### Daily temperature range is positively associated with prevalence of possible sarcopenia in elder people

To investigate the relationship between local temperature fluctuation and prevalence of sarcopenia, we applied China Health and Retirement Longitudinal Study 2013 (CHARLS 2013) database, and enrolled 5737 aged participants (above 60 years old) after screening (Figure S1a). Sarcopenia status was assessed based on the AWGS 2019 algorithm (Figure S1b)[23]. Table 1 demonstrated the baseline demographic characteristics and physical information of included participants. Among 5737 aged people, 29.6% of the total were considered as possible sarcopenia, while 11.1% were sarcopenia. People with sarcopenia were more likely to be older, had decreased BMI index and ASM/Ht$^2$, lower grip strength, worse performance in 5-time chair stand tests, lower gait speed, and more chance to fall, compared with no sarcopenia group. Figure 1a revealed the distribution of prevalence of possible sarcopenia in China. Top 5 provinces with the highest prevalence of possible sarcopenia were Qinghai, Tianjin, Jilin, Gansu, and Hebei.

We also collected diurnal temperature range (DTR) information from 2011.1.1 to 2013.12.31, calculated mean DTR (mDTR) and the proportion of days with DTR above 15 °C in referred period (DTR15%) as the indicators to reflect local temperature variation. Figure 1b demonstrated that southeast China had lower mDTR, including Guangxi, Guangdong, Fujian, Hunan, and Chongqing. Meanwhile, mDTR in northwest China, including Qinghai, Gansu, Xinjiang, and Inner Mongolia, were higher. Figure 1c showed the prevalence of possible sarcopenia and mDTR shared similar distribution patterns in representative cities. Based on these findings, we conducted correlation analysis and found a positive correlation between possible sarcopenia prevalence and mDTR regardless of sex (Total: $R$s = 0.365 $P$ = 3.40E-05, Male: $R$s = 0.274 $P$ = 2.20E-03, Female: $R$s = 0.355 $P$ = 5.50E-05) (Fig. 1d–f). To eliminate the effects of local temperature per se, we used local mean temperature as an adjustment, and discovered that the positive correlation among possible sarcopenia prevalence and mDTR still remained (Total: $R$ = 0.33 $P$ = 2.01E-04, Male: $R$ = 0.237 $P$ = 8.44E-03, Female: $R$ = 0.314 $P$ = 4.20E-04). Similar results were also found in the correlation between possible sarcopenia prevalence and local DTR15% (Total: $R$s = 0.344 $P$ = 9.80E-05, Male: $R$s = 0.262 $P$ = 3.44E-03, Female: $R$s = 0.319 $P$ = 3.22E-04) (Figure S1c–e).

### Exposure to fluctuated temperature aggravates sarcopenia in mice

Next, we attempted to clarify the findings from humans in mouse model. 12-month-old mice were exposed to daily fluctuated temperature (FT) ranging from 10 °C to 25 °C, with 25 °C in the daytime (8:00 am–20:00 pm) and 10 °C in the nighttime (20:00 pm–8:00 am) for 12 weeks. In order to eliminate the effect caused by the temperature per se, we also set two groups with temperatures of 25 °C (Room temperature, RT) and 10 °C (Low temperature, LT) (Fig. 2a). After 12 weeks, we performed rotarod test, grip test, and treadmill test on these groups to evaluate their exercise capacity. FT mice displayed decreased time on rotarod and grid, indicating the dampened coordinate ability and muscle strength, and reduced maximal running time on treadmill, indicating attenuated muscle endurance (Fig. 2b–d). In addition, fluctuated temperature led to reduced mass of quadriceps femoris, tibialis anterior, and gastrocnemius muscles (Fig. 2e–f). Based on the HE staining of gastrocnemius, mean cross-sectional area (CSA) in FT mice was evidently decreased (Fig. 2g–h). Also, FT mice had a significant increase in that of smaller fibers (200–1000 μm$^2$), indicating the shift towards a greater proportion of smaller fibers (Fig. 2h–i). Succinate dehydrogenase (SDH) staining showed a reduced number of oxidative fibers in FT mice (Fig. 2j–k). We also analyzed the expression levels of muscle atrophy-related genes, including *Atrogin1*, *MuRF1*, *Musa1*, and *Myostatin*, and found *Myostatin*, *Atrogin1*, and *MuRF1* were up-regulated in the gastrocnemius of FT mice (Fig. 2l).

Ambient temperature has profound effects on physical status, including metabolism, physical activity, etc. Mice with LT exposure had increased food intake and reduced core temperature (Figure S2a, b). And chronic LT treatment enhanced thermogenesis, revealed by increased energy expenditure (Figure S2c). Under the coexistence of both, the body weight of LT mice did not reveal significantly difference compared with the other two groups (Figure S2d). Besides, insulin tolerance tests (ITTs) and glucose tolerance tests (GTTs) revealed better insulin sensitivity and glucose tolerance performance in LT mice (Figure S2e–h). The level of circulating insulin and HOMA index were similar between LT group and RT group (Figure S2i, j). In FT mice, no significant difference was found in food intake, body weight, rectal temperature, energy expenditure, and other glucose metabolic parameters compared to RT mice (Figure S2a–j). We also monitored their locomotion during 12-week treatment, and found LT mice had decreased physical activity after 4-week exposure (Figure S2k). Meanwhile, FT and RT had a similar level of locomotion (Figure S2k), suggesting that muscle dysfunction in FT mice were not resulted from

**Table 1 | Characteristics of surveyed participants at baseline**

| Characteristic | No sarcopenia | Possible sarcopenia | Sarcopenia |
|---|---|---|---|
| Age | 66.23 ± 5.36 | 68.64 ± 6.63 | 73.44 ± 7.21 |
| Gender | | | |
| Male, *n* (%) | 1838 (54.1%) | 780 (45.9%) | 300 (46.9%) |
| Female, *n* (%) | 1562 (45.9%) | 918 (54.1%) | 339 (53.1%) |
| BMI, kg/m$^2$ | | | |
| Male | 23.18 ± 6.19 | 24.45 ± 9.98 | 18.36 ± 1.74 |
| Female | 24.15 ± 6.35 | 25.3 ± 3.62 | 18.94 ± 1.87 |
| Smoke, *n* (%) | 270 (7.9%) | 153 (9.0%) | 52 (8.1%) |
| Alcohol, *n* (%) | 1260(37.06%) | 436(25.68%) | 175(27.39%) |
| ASM/Ht$^2$, kg/m$^2$ | | | |
| Male | 7.54 ± 0.94 | 7.7 ± 1.19 | 6.45 ± 0.58 |
| Female | 5.84 ± 1.03 | 6.06 ± 0.72 | 4.55 ± 0.51 |
| Handgrip, kg | | | |
| Male | 38.15 ± 6.41 | 30.96 ± 8.21 | 26.75 ± 7.17 |
| Female | 26.44 ± 5.93 | 21.4 ± 6.61 | 17.67 ± 5.73 |
| 5-time chair stand test, s | 8.96 ± 1.75 | 14.68 ± 5.04 | 14.53 ± 5.16 |
| Gait speed, m/s | 0.75 ± 0.24 | 0.62 ± 0.3 | 0.58 ± 0.18 |
| Fall, *n* (%) | 531(15.6%) | 357(21.0%) | 133(20.8%) |

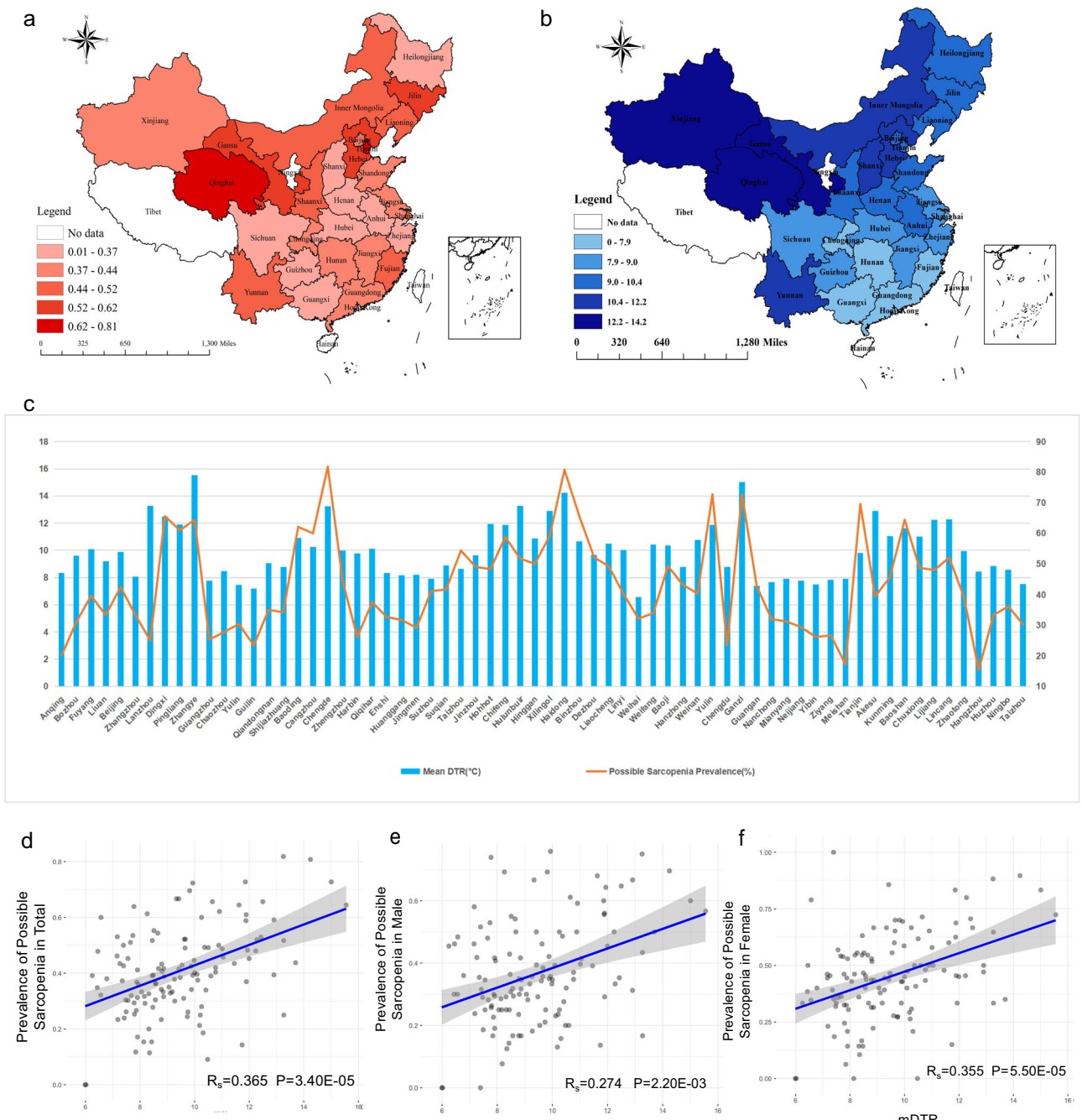

**Fig. 1 | Daily temperature range is positively associated with the prevalence of possible sarcopenia in elder people. a** Geographical distributions of the prevalence of possible sarcopenia in China based on CHARLS database 2013. No prevalence data in Tibet, Ningxia, Hongkong, Macao, and Taiwan. **b** Geographical distributions of local mean DTR (mDTR) ranging from 2011 to 2013. mDTR of each province were the average value of DTR in the included cities. **c** Bar plot of prevalence of possible sarcopenia and mDTR in representative cities. **d–f** Scatter plot of correlation between the prevalence of possible sarcopenia in 124 cities and their mDTR. Data presented are mean ± s.d. Spearman correlation analysis was conducted. Source data are provided as a Source Data File.

a lack of exercise. To further investigate their stress status, we detected the level of circulating stress hormones after 12 weeks, including epinephrine, cortisol, and vasopressin, and found no significant difference among these groups (Figure S2l–n).

These findings show that prolonged exposure to fluctuated temperature promotes muscle atrophy and dampens muscle function, leading to the aging-like phenotype in muscle of mid-aged mice.

**Fluctuated temperature alters the microbiota composition**

Ambient temperature profoundly impacts gut microbiota[24]. Thus, we performed 16 S ribosomal DNA analysis of microbiota in feces from

RT, LT, and FT mice. We first assessed the gut microbial community structure by analyzing the α- and β-diversity. α-diversity was reflected by Chao1 and Shannon's index, indicating the community richness and evenness. And no significant difference was found in Chao1 index among these groups, while the LT group had higher level of Shannon's index, suggesting that cold exposure might lead to an even distribution in abundance of the bacterial species (Fig. 3a, b). Principal component analysis (PCA), representing β-diversity, revealed that the composition and abundance of the microbiota in RT, LT, and FT mice were significantly different from each other (Fig. 3c). Figure 3d revealed the distribution of gut microbiota at family level of RT, LT, and FT mice. The

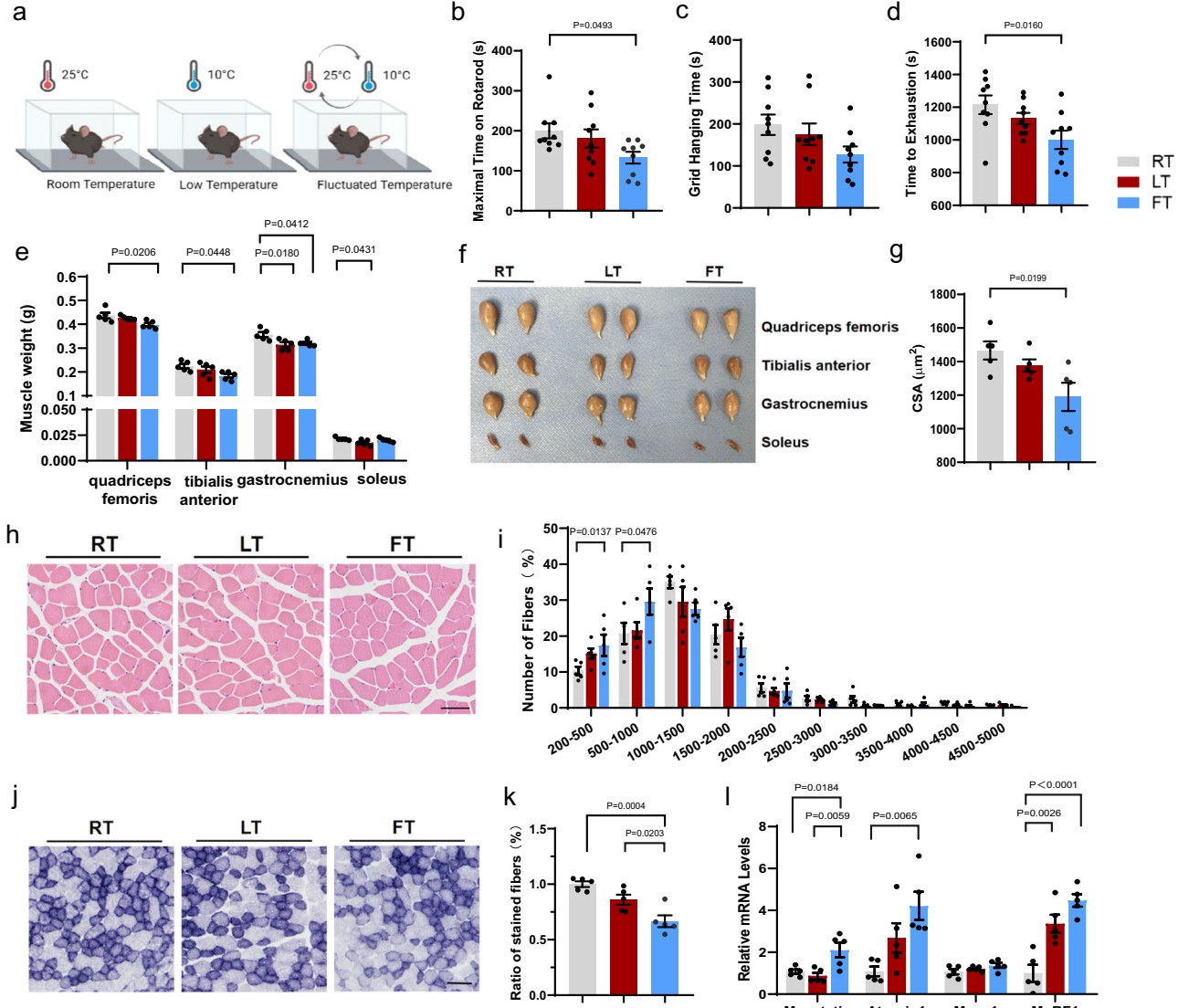

**Fig. 2 | Exposure to fluctuated temperature aggravates sarcopenia in mice.**
**a** Experimental scheme. 12-month-old male mice were exposed to room tempera-
ture (RT), low temperature (LT) and fluctuated temperature (FT) for 12 weeks.
**b–d** Determination of exercise capacity by conducting rotarod test (**b**), grip test (**c**)
and treadmill test (**d**). *n* = 9 biologically independent animals per group. **e**, **f** Weight
and representative pictures of dissected quadriceps femoris, tibialis anterior, gas-
trocnemius and soleus muscles. *n* = 9 biologically independent animals per group.
**g–i** Representative images of HE staining of gastrocnemius muscle cross-section
(**h**), quantification of mean cross-sectional areas (**g**) and distribution of muscle

fibers (**i**). Bar = 80 μm. *n* = 5 biologically independent animals per group.
**j**, **k** Representative images of succinate dehydrogenase (SDH) staining of tibialis
anterior muscle cross-section (**j**) and quantification (**k**) of stained muscle fibers.
Bar = 125 μm. *n* = 5 biologically independent animals per group. **l** Expression levels
of *Myostatin*, *Atrogin1*, *Musa*, and *MuRF1* measured by qPCR. *n* = 5 biologically
independent animals per group. Data presented are mean ± s.e.m. One-way ANOVA
test for multiple comparisons with Tukey's test for post hoc corrections. Source
data are provided as a Source Data File.

top 5 abundant bacteria at family level were *Muribaculaceae, Lachnos-
piraceae, Prevotellaceae, Ruminococcaceae,* and *Rikenellaceae. Rumino-
coccaceae* was reported to accumulate in intestinal tract after cold
exposure[25]. Previous study revealed that *Muribaculaceae* were sig-
nificantly enhanced in warm-induced mice[26], and *Muribaculaceae* was
decreased in LT group, indicating they could be susceptible to ambient
temperature. Hierarchical clustering of each genus also confirmed
the unique effect brought from fluctuated temperature on the gut
microbiota (Fig. 3e). Furthermore, we conducted shotgun metage-
nomic sequencing on these feces materials. Linear discriminant analysis
effect size (LEfSe) analysis demonstrated the characterized
bacteria in each group (Fig. 3f). Gut microbiota in FT group featured
with increase of *Parabacteroides_distasonis, Duncaniella_dubosii, Para-
sutterella_sp._NM82_D38, Clostridium_sp._CAG:557, Bacteroides_sp._224,
Prevotella_ihumii*. We also noticed the gut microbiota decreased in FT

group, including *Candidatus_Amulumruptor, Roseburia, Eubacterium,
Enterocloster, Eisenbergiella,* and *Kineothrix* (Fig. 3g).

These data suggest that fluctuated temperature impacts micro-
biota composition, and whether it is involved in the process of sar-
copenia needs further exploration.

## Transplantation of microbiota from fluctuated temperature-treated mice dampens muscle function

To test whether fluctuated temperature-induced muscle dysfunction
was mediated by altered gut microbiota, we performed fecal micro-
biota transplantation (FMT) experiment. 12-month-old mice were
divided into three groups and treated with fecal materials from RT
(FMTrt), LT(FMTlt), and FT (FMTft) mice, respectively. After 8 weeks,
compared to FMTrt mice, FMTft mice had dampened muscle function,
indicated by weakened coordination, decreased muscle strength and

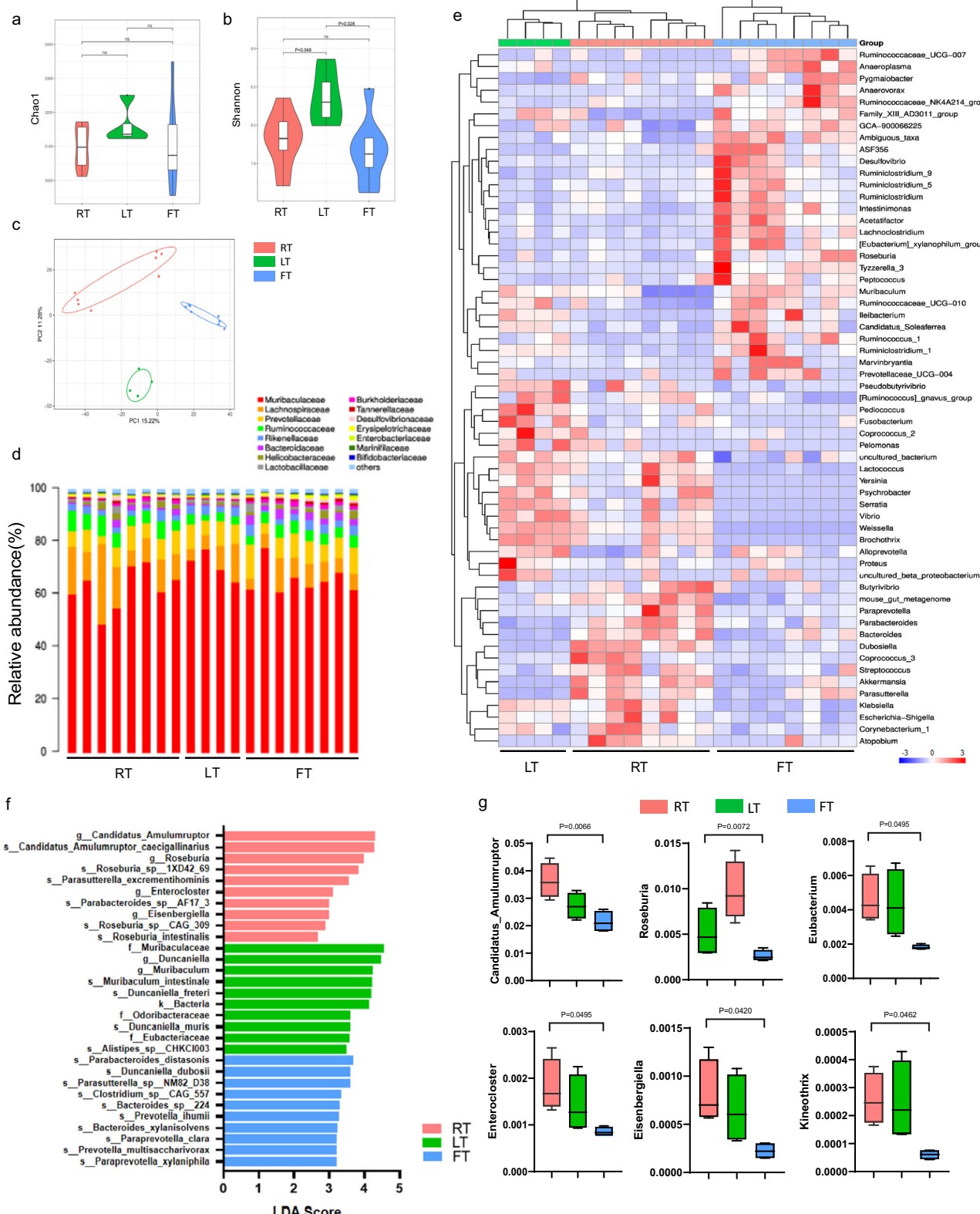

**Fig. 3 | Fluctuated temperature alters the microbiota composition.**
**a**, **b** Shannon diversity and Chao1 of 16 S rDNA sequencing on fecal microbiota from mice exposed to RT (*n* = 8 biologically independent animals per group), LT (*n* = 4 biologically independent animals per group) and FT (*n* = 8 biologically independent animals per group). **c** PCA based on 16 S rDNA sequencing. **d** Bar chart of the relative microbiome abundance at family level. **e** Hierarchically clustered heatmap of differentiated distributed microbiota (*P* <0.05) at genus level. **f** Microbiota with top 10 LDA score in each group based on LEfSe analysis. Log10 (LDA score) were shown. **g** Decreased genera in FT group. *n* = 4 biologically independent animals per group. Data presented are mean±s.e.m. One-way ANOVA test for multiple comparisons with Tukey's test for post hoc corrections. Source data are provided as a Source Data File.

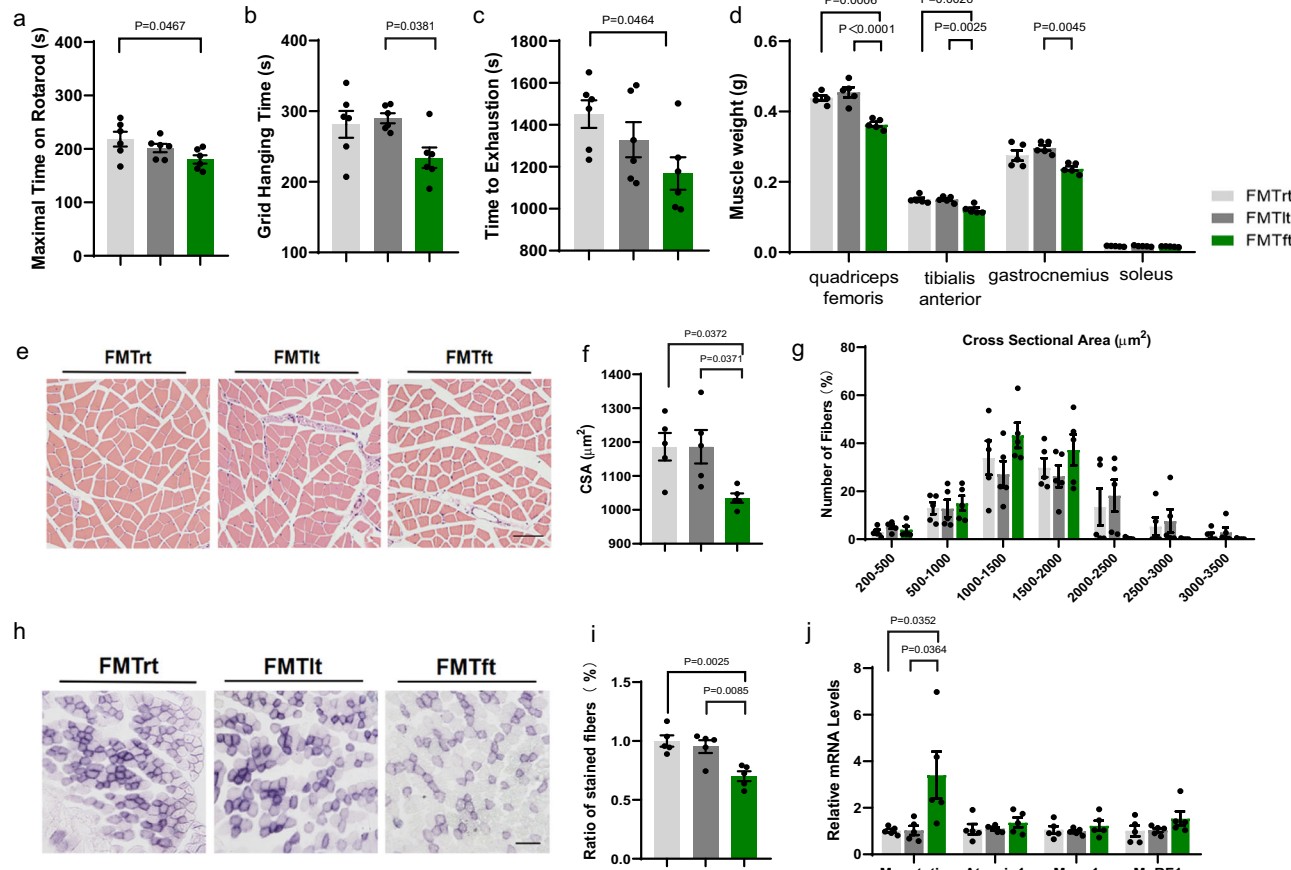

**Fig. 4 | Transplantation of microbiota from fluctuated temperature-treated mice dampens muscle function. a–c** Determination of exercise capacity by conducting rotarod test (**a**), grip test (**b**), and treadmill test (**c**) in FMTrt, FMTlt, and FMTft mice. $n = 6$ biologically independent animals per group. **d** Weight of dissected quadriceps femoris, tibialis anterior, gastrocnemius, and soleus muscles. $n = 5$ biologically independent animals per group. **e–g** Representative images of HE staining of cross-section of gastrocnemius muscle (**e**), quantification of mean cross-sectional area (**f**), and distribution of muscle fibers (**g**). Bar = 80 µm. $n = 5$ biologically independent animals per group. **h–i** Representative images of succinate dehydrogenase (SDH) staining of cross-section of tibialis anterior (**h**) and quantification (**i**) of stained muscle fibers. Bar = 125 µm. $n = 5$ biologically independent animals per group. **j** Expression levels of *Myostatin*, *Atrogin1*, *Musa*, and *MuRF1* measured by qPCR. $n = 5$ biologically independent animals per group. Data presented are mean±s.e.m. One-way ANOVA test for multiple comparisons with Tukey's test for post hoc corrections. Source data are provided as a Source Data File.

endurance (Fig. 4a–c). Evidences supporting muscle atrophy were also observed in FMTft mice, including decreased muscle mass (Fig. 4d), mean CSA, and large muscle fiber based on HE staining (Fig. 4e–g). The number of oxidative fibers was decreased in the tibialis anterior muscle of FMTft, compared with FMTrt and FMTlt mice (Fig. 4h–i). Moreover, the mRNA level of *Myostatin* was higher in gastrocnemius of FMTft mice (Fig. 4j).

These findings demonstrate that altered gut microbiota induced by fluctuated temperature contributes to muscle atrophy and dysfunction.

## Fluctuated temperature treatment leads to mitochondrial dysfunction in muscle

Damaged muscle function and decreased oxidative muscle fiber in FT mice led us to wonder whether the mitochondrial function was affected. To test this, we detected the morphology and function of mitochondria in FT and FMTft mice. Transmission electron microscopy analysis showed decreased mitochondrial size in gastrocnemius of FT (Fig. 5a, b). Since mitochondria serve as core workplace for ATP generation and ROS production, we further examined ATP contents and ROS levels. FT treatment led to decreased ATP contents and elevated ROS levels in gastrocnemius, and these effects were also found in FMTft mice (Fig. 5c, d). We further examined mitochondrial membrane potential ($\Delta\psi_M$), an

important indicator to evaluate mitochondrial status, loss of which caused insufficient ATP production and triggered cell apoptosis[27]. Through JC-1 probe, we found increased JC-1 monomers level in isolated mitochondria of FT mice, suggesting the decrease of $\Delta\psi_M$ (Fig. 5e). We also used TMRE probe to confirm $\Delta\psi_M$ level, and found decreased TMRE fluorescence in muscle of FT and FMTft mice (Fig. 5f). Mitochondria were semi-autonomous organelles which had their own DNA (mtDNA)[28], we further examined mtDNA and found a higher ratio of mitochondrial DNA to nuclear DNA (mtDNA/nuDNA) in the gastrocnemius of FT mice, indicating that there might be an increase in wasted mitochondria after FT treatment (Fig. 5g). These results confirm our assumption that fluctuated temperature dampens mitochondrial function in muscular tissue.

The mitochondrial biogenesis, dynamics, and selective degradation (mitophagy) ensure the maintenance of well-functioning mitochondria[29]. To further investigate the contributing factor of mitochondrial dysfunction, we analyzed the related genes by qRT-PCR. No significant difference was found in genes involved in mitochondrial biogenesis and mitochondrial dynamics (Fig. 5h). However, genes related to autophagy and mitophagy, including *Beclin1*, *Atg5*, *Pik3c3*, and *Park2*, were significantly decreased in the gastrocnemius of FT mice (Fig. 5h). We also investigated these markers in protein level. Protein levels of TFAM, PGC1α, (biogenesis), and MFN1, OPA1 (dynamics) showed no significant difference between

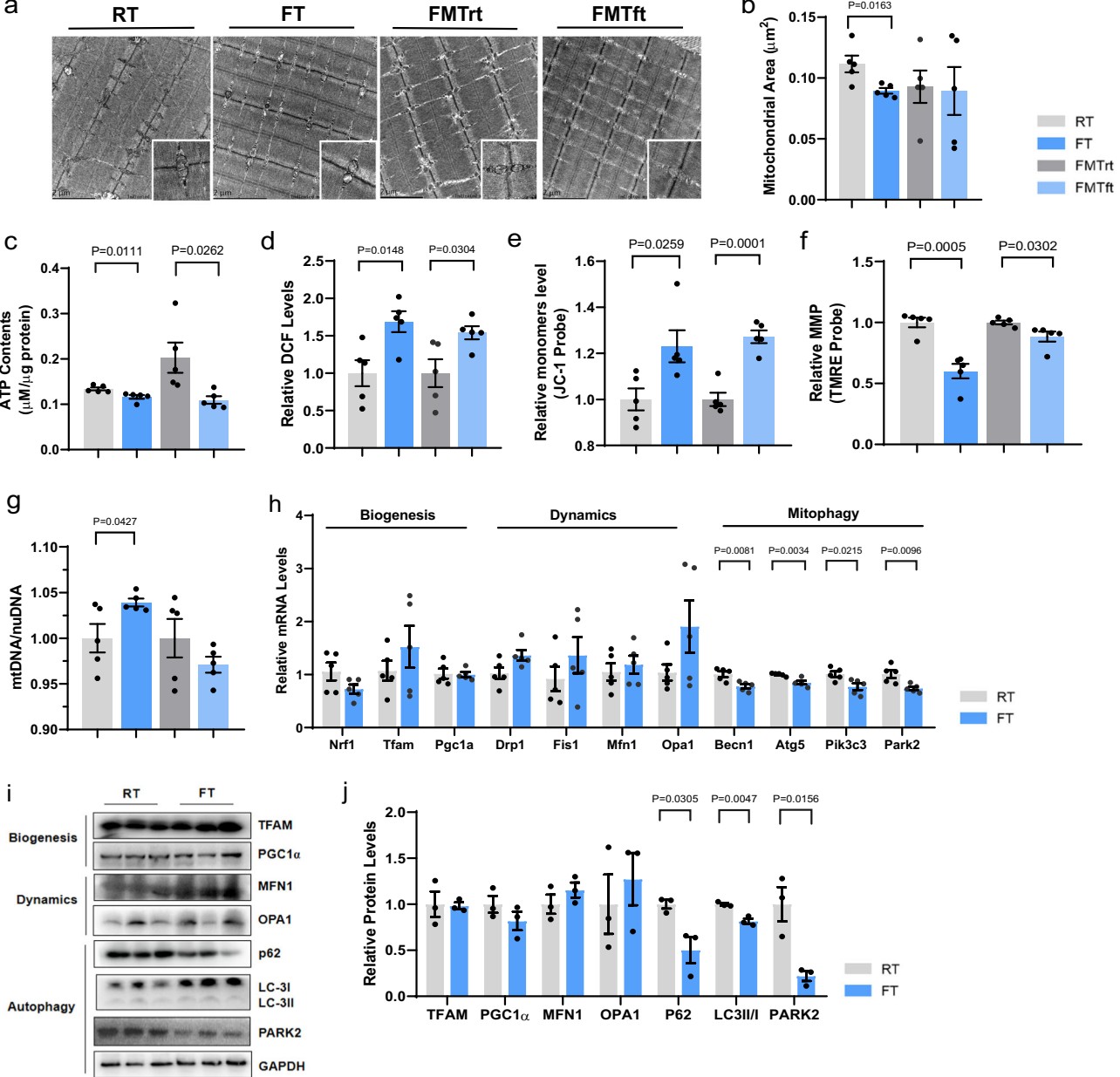

Fig. 5 | Fluctuated temperature treatment leads to mitochondrial dysfunction in muscle. a, b Representative transmission electron micrographs (TEM) of intermyofibrillar (IMF) mitochondria from gastrocnemius of RT, FT, FMTrt, and FMTft mice (a) and quantification (b) of relative mitochondria area. $n = 5$ biologically independent animals per group. c Determination of ATP contents of muscle tissue, protein levels were used for adjustment. $n = 5$ biologically independent animals per group. d Determination of ROS level in digested muscle cells through DCF probe. $n = 5$ biologically independent animals per group. e, f Determination of mitochondrial membrane potential (MMP) by JC-1 probe and TMRE probe in isolated mitochondria from muscle tissue. $n = 5$ biologically independent animals per group. g Quantification of mtDNA/nuDNA ratio. $n = 5$ biologically independent animals per group. h Expression levels of mitochondrial biogenesis-, dynamics-, and mitophagy-related genes by qPCR. $n = 5$ biologically independent animals per group. i, j Western blot (i) of mitochondrial biogenesis-, dynamics-, and mitophagy-related biomarkers and quantification of the bands in the gels (j). $n = 3$ biologically independent animals per group. Data presented are mean ± s.e.m. One-way ANOVA test for multiple comparisons with Tukey's test for post hoc corrections. Source data are provided as a Source Data File.

muscle of FT and RT mice (Fig. 5i, j). Protein levels related to autophagy and mitophagy, including P62, LC3II/I, and PARK2 were decreased in FT mice (Fig. 5i, j). Besides, we further detected whether other biological process was affected in FT mouse model, including protein synthesis, degradation, and inflammation. No significant difference was found in the protein level of phospho/total-mTOR, Ubiquitin, IL6, and TNFα between RT and FT group (Figure S3a). Similar results were also found in FMT mouse model (Figure S3b). Noticeably, FMTft mice also had decreased levels of P62, LC3II/I, and PARK2 (Figure S3b).

These findings demonstrate that exposure to fluctuated temperature damages mitochondrial function in muscle accompanied by inhibited autophagy- and mitophagy-related makers.

### Lysine degradation metabolite, aminoadipic acid, mediates fluctuated temperature-induced sarcopenia

To elucidate the link between altered microbiota composition and muscle mitochondrial dysfunction, we hypothesized that there might be a changed metabolite contributing to these effects. Based on the KEGG analysis of shotgun metagenomic sequencing, several identified

pathways were differentially enhanced in FT mice. Among the top 10 significant pathways in FT group (ranked by LDA score), the only metabolism-related pathway, the lysine degradation pathway, aroused our attention (Fig. 6a). To test whether degraded lysine was involved in the fluctuated temperature-related muscle atrophy, we investigated the levels of lysine and its degradation product, aminoadipic acid (α-AA) in circulation. No significant difference was found in the level of lysine; however, circulating α-AA level was increased in FT mice, compared to RT mice (Fig. 6b, c). α-AA was reported to be a glutamine synthetase inhibitor[30], and a potential regulator in glucose homeostasis[31,32]. However, whether increased α-AA level affects muscle function needs further investigation.

To verify this hypothesis, we treated 12-month-old mice with α-AA via drinking water. After 4 weeks, mice administered with α-AA had a similar phenotype as mice exposed to fluctuated temperature, revealed by decreased capacity of muscle coordination, endurance, and strength (Fig. 6d–f). Muscle mass of quadriceps femoris was significantly reduced in α-AA-treated mice (Fig. 6g, h), along with lower muscle fiber area and oxidative muscle fiber number (Fig. 6i–m). Since fat infiltration was also a typical characteristic of sarcopenia, we stained muscle tissue with BODIPY dye, and noticed the increase of stained lipid droplet in AA-treated muscle (Fig. 6n, o). The mRNA levels of *Atrogin1*, *Musa1*, and *MuRF1* were elevated after α-AA administration (Fig. 6p). Additionally, reduced ATP contents and $\Delta\psi_M$ were also found in gastrocnemius of α-AA-treated mice (Fig. 6q, r). Multiple protein markers were analyzed by Western blot. In consistent with FT mice, mice with α-AA treatment had decreased p62/SQSTM, LC3II/I ratio, and PARK2 levels (Figure S3c). The blockage of autophagic flux after α-AA treatment was revealed using lysosomal inhibitor chloroquine (CQ), which led to reduced level of LC3II (Figure S3d). And mice with both α-AA and CQ administration had lower level of autophagy markers compared with control group with CQ treatment alone (Figure S3d).

Together, these findings show α-AA supplements can mimic the fluctuated temperature-induced phenotype, suggesting that elevated circulating a-AA might be a crucial link between altered microbiota composition and impaired mitochondrial function in muscle tissue.

## Aminoadipic acid damages mitochondrial function through inhibiting mitophagy

Mitophagy is a process which lysosome selectively removes dysfunctional mitochondria to maintain cellular balance. Decreased mitophagy was found during aging, which destructed myofiber homeostasis, ultimately leading to muscle atrophy. Based on the decreased level of autophagy- and mitophagy- related markers in muscle of FT-treated mice, we further investigated whether α-AA supplement directly led to decreased mitophagy in C2C12 cell line. Differentiated C2C12 myotubes with α-AA supplementation had a dose-dependent change of biomarkers for autophagy, with decrease of the ratio of LC3II/I and reduction of p62/SQSTM level (Fig. 7a). Consistently, isolated mitochondria fragment from α-AA-treated myotubes also had decreased ubiquitination and p62/SQSTM enrichment, suggesting the inhibited mitophagy level caused by α-AA (Fig. 7a). Besides, the mRNA levels of autophagy- and mitophagy- related genes, including *Atg5*, *Lamp*, *Pik3c3*, *Atg8,* and *Park2*, were decreased in C2C12 myotubes treated with α-AA (Fig. 7b), compared with the controls. Furthermore, to inhibit autophagy flux, C2C12 cells treated with GFP-LC3 adenovirus and Mitotracker were exposed to CQ. Co-staining of red fluorescence (mitochondria) and green fluorescence (autophagosomes) were lower in cells with α-AA/CQ treatment, compared with cells with DMSO/CQ treatment, indicating the decreased mitophagy flux led by α-AA supplement (Fig. 7c, d). Ratio of mtDNA/nuDNA was increased after 100 μM α-AA treatment, and decreased when concentration reaching 200 μM (Fig. 7e). Enhanced mitochondrial abundance were also found in Mitotracker-stained C2C12 after 100 μM α-AA treatment (Figure S4a).

We also evaluated mitochondrial function in vitro. In C2C12 myotubes, α-AA treatment led to increased ROS level (Figure S4b), revealed by DCF fluorescence, and reduced $\Delta\psi_M$ (Figure S4c, d), revealed by both JC-1 and TMRE probe, and decreased ATP content (Fig. 7g). To inquire whether restoring mitophagy level could improve mitochondrial dysfunction led by α-AA exposure, C2C12 myotubes were treated with Park2-plasmid, which partly enhanced the level of mitophagy (Fig. 7f). α-AA-induced mitochondrial dysfunction were partly blunted by Park2 overexpression, as revealed by restored levels of ATP content and TMRE level, along with decreased ROS level (Fig. 7g–i). These observations demonstrate that α-AA can exert its detrimental effects by inhibiting mitophagy in vitro, and enhancing mitophagy level can partly reverse the effects.

## *Eubacterium* supplementation alleviates the detrimental effects induced by fluctuated temperature

The beneficial effects of probiotic supplementation prompted us to inquire whether transplantation of certain bacteria could counteract the detrimental outcome brought from changed temperature. We searched the decreased bacteria in feces of FT mice from gut metagenomic profiles, and chose *Eubacterium* for the reason that (1) The level *Eubacterium* was negatively associated with the bacteria had high LDA score in FT-treated mice, including *Bacteroides*, *Paraprevotella* (Fig. 2f, Figure S5a); (2) Decreased *Eubacterium* level in FT group was partly contributing to the Lysine degradation pathway based on metagenomic results (Figure S5b). Therefore, we further inquired whether *Eubacterium* supplement could improve the muscle dysfunction led by fluctuated temperature.

12-month-old mice were placed into the forementioned chambers with fluctuated temperature for 4 weeks, following oral administration of *Eubacterium* twice a week for other 8 weeks. Mice with *Eubacterium* supplement had better performance in exercise capacity tests, with extended rotarod and grip time (Fig. 8a–c). *Eubacterium* transplantation decelerated the fluctuated temperature-induced muscle atrophy, demonstrated by enhanced muscle mass, mean myofiber area, and increased large muscle fiber (Fig. 8d–h). SDH staining showed a relative enhancement in the number of oxidative fibers after *Eubacterium* supplement (Fig. 8i, j). Among muscle atrophy-related genes, *Myostatin* revealed a decreased tendency in gastrocnemius of *Eubacterium*-treated mice (Fig. 8k). Furthermore, Western blot showed that *Eubacterium* treatment enhanced LC3II/I ratio, P62, and PARK2 expression in muscle tissue (Figure S3e).

## Discussion

In this study, we initiated with the positive correlation between DTR and prevalence of possible sarcopenia based on Chinese database, and proved the causal relationship of fluctuated temperature (FT) and sarcopenia in mouse model. Altered gut microbiota mediated this process, and enhanced circulating aminoadipic acid level, therefore leading to mitochondrial dysfunction in muscle tissue. Supplementing *Eubacterium*, a bacteria strain decreased in feces of FT mice, alleviated muscle atrophy induced by FT exposure.

Increased DTR was reported to be a risk factor of many diseases based on epidemiological data[3–5]. Tang et al. reported a significantly positive relationship between long-term DTR exposure and all-cause mortality, with a 13% higher risk of all-cause mortality based on a 1 °C increase in the DTR exposure[33]. In our study, we chose 10–25 °C for the mouse model, based on (1) the maximum mean DTR in all the cities included were 15.55 °C; (2) the mean daily maximum temperature is 21.6 °C, and the minimum is 11.9 °C. The set of 25 °C in daytime and 10 °C in night time were more appropriate to mimic the effects of daily temperature change on Chinese.

Multiple factors contribute to muscle atrophy in later stage of life, especially decreased physical activity. We investigated whether muscle dysfunction led by fluctuated temperature was affected by physical

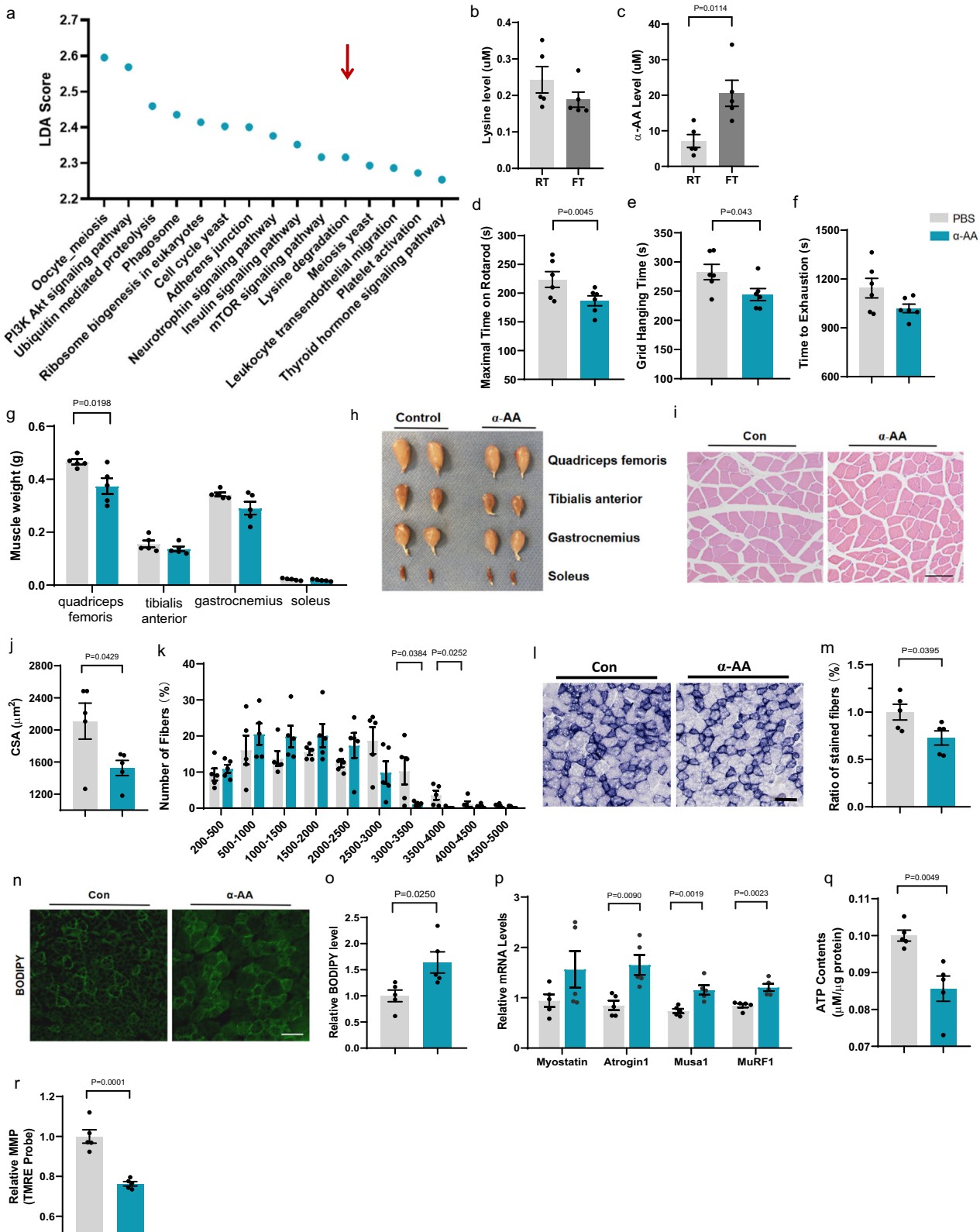

inactivity, and found no significant correlation between mDTR and physical activity of local people based on CHARLS database (See point-by-point response). In mouse model, RT and FT mice had similar level in physical activity, which excluded the possibility that reduced physical activity participate in the FT-induced muscle dysfunction.

Many bacteria were reported to be associated with sarcopenia[34]. Van Tongeren et al. have reported that the increased level of

*Enterobacteriacae*, and the decreased level of *Lactobacilli, Bacterioides/ Prevotella*, and *Faecalibacterium prausnitzii* in elder people were related with high frailty scores[35]. And Picca et al. showed that fecal microbiota from older individuals with physical fragility and sarcopenia were characterized by increased *Dialister, Pyramidobacter*, and *Eggerthella*, and decreased *Slackia* and *Eubacterium*[36]. To overcome dampened muscle function led by FT treatment, we selected the *Eubacterium*, a

**Fig. 6 | Lysine degradation metabolite, aminoadipic acid, mediates fluctuated temperature-induced sarcopenia. a** LEfSe analysis on KEGG pathway of gut microbiome from FT group. **b, c** Quantification of circulating lysine and α-AA. *n* = 5 biologically independent animals per group. **d–f** Determination of exercise capacity by conducting rotarod test (**d**), grip test (**e**), and treadmill test (**f**) in mice treated with α-AA and drinking water. *n* = 6 biologically independent animals per group. **g, h** Weight and representative pictures of dissected quadriceps femoris, tibialis anterior, gastrocnemius, and soleus. *n* = 5 biologically independent animals per group. **i–k** Representative images of HE staining of cross-section of gastrocnemius muscle (**i**), quantification of mean cross-sectional area (**j**), and distribution of muscle fibers (**k**). Bar = 80 μm. *n* = 5 biologically independent animals per group.

**l, m** Representative images of succinate dehydrogenase (SDH) staining of cross-section of tibialis anterior (**l**) and quantification (**m**) of stained muscle fibers. Bar = 125 μm. *n* = 5 biologically independent animals per group. **n–o** Representative images of BODIPY staining of cross-section of tibialis anterior (**n**) and quantification (**o**) of the stained area. Bar = 125 μm. *n* = 5 biologically independent animals per group. **p** Expression levels of *Myostatin, Atrogin1, Musa,* and *MuRF1* measured by qPCR. *n* = 5 biologically independent animals per group. **q–r** Determination of ATP contents in muscle tissue and mitochondrial membrane potential (TMRE probe) in isolated mitochondria. *n* = 5 biologically independent animals per group. Data presented are mean±s.e.m. Two-tailed unpaired *t* test for binary comparison. Source data are provided as a Source Data File.

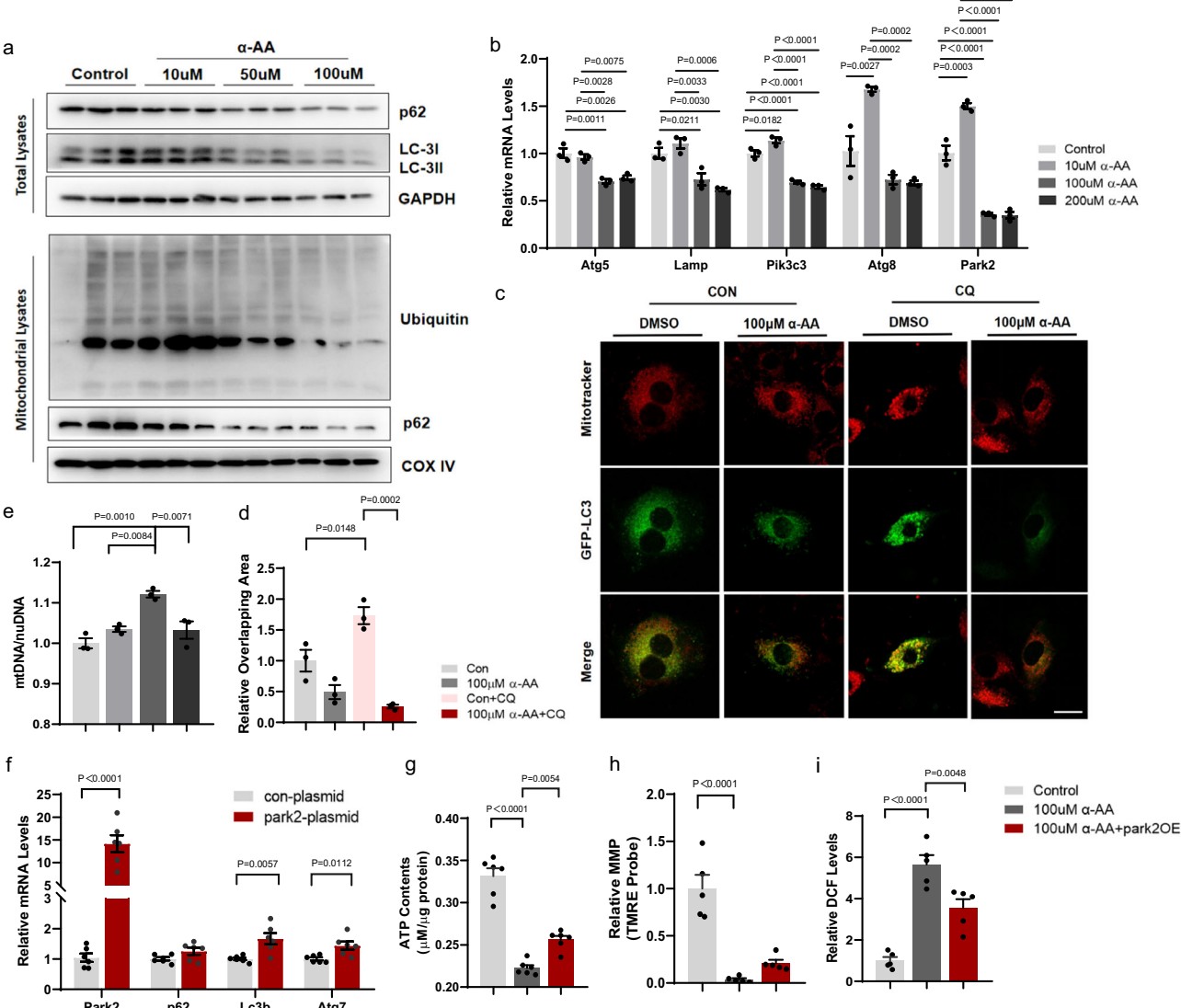

**Fig. 7 | Aminoadipic acid damages mitochondrial function by inhibiting mitophagy. a** Western blot of autophagy- and mitophagy-related proteins in total lysate and mitochondrial fraction in C2C12 cells. *n* = 3 biologically independent cells per group. **b** Expression levels of autophagy- and mitophagy-related genes by qPCR. *n* = 3 biologically independent cells per group. **c, d** Representative images of mitophagy flux by co-staining of mitochondria (Red, Mitotracker) and LC3 (green, GFP-LC3) after 6-hr CQ treatment (**d**) and quantification (**e**) of the overlapping area. Bar = 10 μm. *n* = 3 biologically independent cells per group. **e** Quantification of

mtDNA/nuDNA ratio by qPCR. *n* = 3 biologically independent cells per group. **f** Expression levels of autophagy- and mitophagy-related genes in C2C12 after park2-plasmid transfection by qPCR. *n* = 6 biologically independent cells per group. **g–i** Determination of ATP contents, MMP, ROS in α-AA-treated C2C12 with park2-plasmid transfection. *n* = 5 biologically independent cells per group. Data presented are mean ± s.e.m. One-way ANOVA test for multiple comparisons with Tukey's test for post hoc corrections. Source data are provided as a Source Data File.

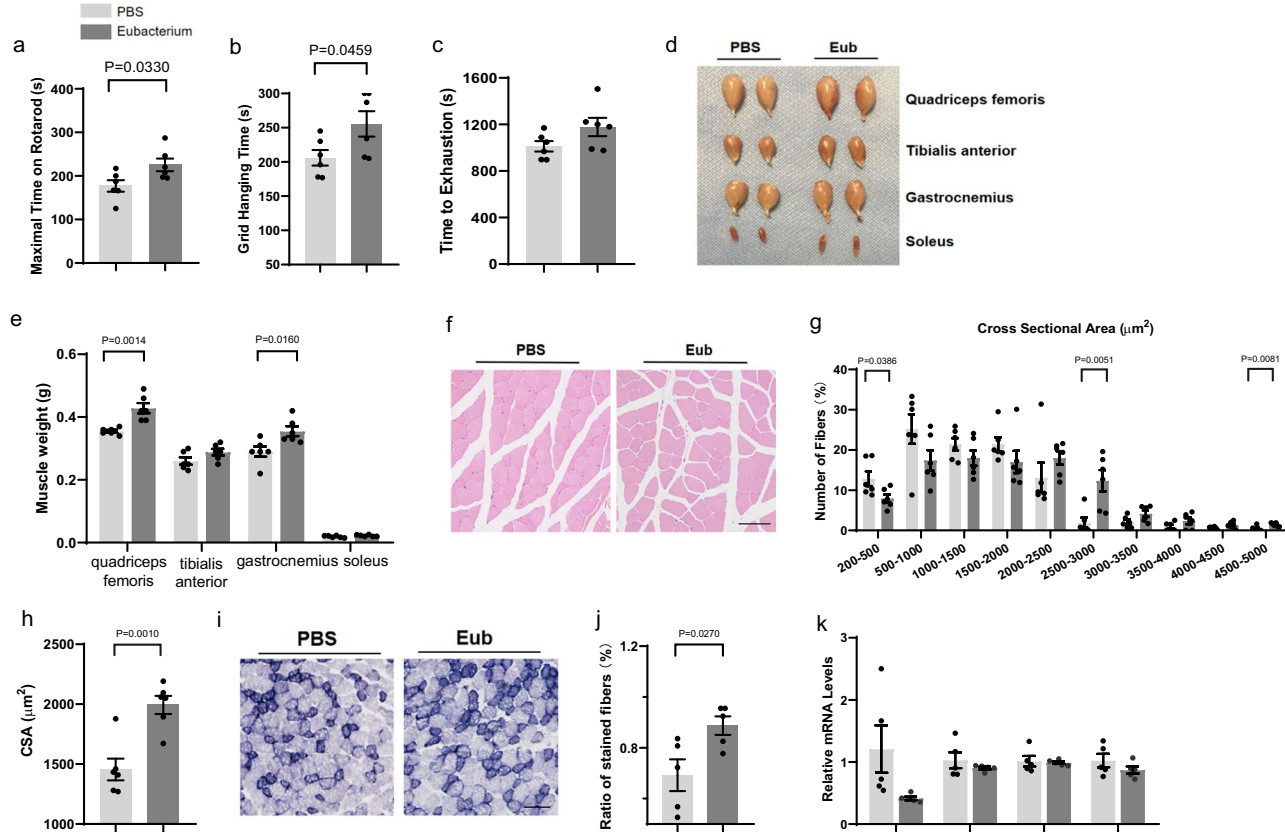

**Fig. 8 | *Eubacterium* supplementation alleviates the detrimental effects induced by fluctuated temperature. a–c** Determination of exercise capacity by conducting rotarod test (**a**), grip test (**b**), and treadmill test (**c**) treated with *Eubacterium* and PBS. $n = 6$ biologically independent animals per group. **d, e** Representative pictures and weight of dissected quadriceps femoris, tibialis anterior, gastrocnemius, and soleus muscles. $n = 6$ biologically independent animals per group. **f–h** Representative images of HE staining of cross-section of gastrocnemius muscle (**f**), distribution of muscle fibers (**g**), and quantification of mean cross-sectional area (**h**). Bar = 80 μm. $n = 6$ biologically independent animals per group. **i, j** Representative images of succinate dehydrogenase (SDH) staining of cross-section of tibialis anterior muscles (**i**) and quantification (**j**) of stained muscle fibers. Bar = 125 μm. $n = 6$ biologically independent animals per group. **k** Expression levels of *Myostatin*, *Atrogin1*, *Musa*, and *MuRF1* measured by qPCR. $n = 5$ biologically independent animals per group. Data presented are mean±s.e.m. Two-tailed unpaired *t* test for binary comparison. Source data are provided as a Source Data File.

decreased bacteria in FT group. *Eubacterium* was reported to be enriched in individuals with exercise habits, and decreased in elder people with physical frailty and sarcopenia[36–38]. Here, based on the correlation analysis (Figure S5a), we found *Eubacterium* were negatively correlated with the bacteria that had high LDA score in FT-treated mice, including *Bacteroides* and *Paraprevotella* and positively correlated with the beneficial bacteria on muscle function, such as *Ruminococcus*[39]. *Eubacterium* supplement significantly increased muscle mass, including quadriceps femoris and gastrocnemius (GA). The mass of tibialis anterior (TA) was not elevated in a statistically different way, suggesting that reversing the decreased *Eubacterium* level alone can not entirely restore the muscle dysfunction caused by FT treatment. As listed in Figure S5c, the bacteria species pertaining to *Lachnospiraceae* and *Muribaculaceae* also contributed to the altered KEGG pathway in FT group, indicating other bacteria might also participate in the FT-related sarcopenia.

Accumulation of dysfunctional mitochondria contributes to sarcopenia. In order to maintain a healthy mitochondrial pool, mitophagy is indispensable in skeletal muscle cell[29]. Impaired mitophagy levels were found during aging, with reduced Parkin contents[40,41]. Decreased mitophagy exacerbated cellular dysfunction[42], while restoring mitophagy levels by natural compound improved exercise capacity in aged rodents[43]. However, some studies revealed the opposite results that mitophagy flux elevated in aging muscle, resulting in reduced mitochondrial content[29]. Though the conflicting results exists, the general consensus is that insufficient and overactivated mitophagy are both detrimental to cellular homeostasis. Here, we found decreased autophagy- and mitophagy-related markers in the muscle of FT, FMTft and α-AA-treated mice, and reversing mitophagy level can partly improve mitochondrial dysfunction.

Together, our findings reveal the contributing effects of high DTR on sarcopenia, and provide novel perspectives on the gut-muscle axis.

## Methods
### Human data analysis
We collected human data from China Health and Retirement Longitudinal Study (CHARLS, http://charls.pku.edu.cn/en), which is a nationwide database regarding lifestyle habits and health-related information among people over 45 years old. Ethical approval for all the CHARLS waves was granted by the Institutional Review Board at Peking University, and all participants signed informed consent.

In this study, CHARLS 2013 was used for analysis, because participants in this year had comprehensive location information. 5737 out of 18,612 participants were recruited after excluding participants failed to meet our criterion (Figure S1a). Diagnostic algorithm of sarcopenia was based on the AWGS 2019 (Figure S1b)[23]. In short, people with low muscle strength with or without reduced physical performance are diagnosed as possible sarcopenia. People with low muscle mass, together with low muscle strength or low physical performance are diagnosed as sarcopenia. When someone has low muscle mass, low muscle strength, and low physical performance, he/she will be

recognized as severe sarcopenia. For participants who have none of those symptoms are referred as no sarcopenia. And the criterion of 'low muscle strength', 'low physical performance' and 'low muscle mass' are all based on AWGS 2019. According to AWGS 2019, the criterion of low muscle mass is based on ASM/Ht$^2$. And appendicular skeletal muscle mass (ASM) is calculated using the anthropometric equation based on Chinese population: $ASM = 0.193 \times$ body weight $+ 0.107 \times$ height $- 4.157 \times$ sex $- 0.037 \times$ age $- 2.631$. Sex value is assigned to 1 in male, 2 in female. The cutoff for defining low mass is the lowest 20th% percentile of ASM/Ht$^2$ in study population, that is <5.04 kg/m$^2$ in women and <6.85 kg/m$^2$ in men.

Meteorological information were collected from Chinese Research Data Services (CRDS). 124 cities searching for meteorological data were based on the origin of participants. Diurnal temperature range (DTR) was used as the indicator to reflect the daily temperature stability. Since 15 °C are the maximal mean DTR in these cities, we used 1) the proportion of days with DTR above 15 °C in 2011.1.1-2013.12.31 (DTR15%), 2) mean DTR of these years (mDTR) to reflect local temperature variation.

### Animals
Animal experiments were approved by the Animal Care and Use Committees of the Laboratory Animal Research Center at Xiangya Medical School of Central South University. All experiments were performed on 12-month-old C57BL/6 J mice, obtained from SLAC Laboratory Animals Co. Ltd (China). Except for the mouse model of fluctuated temperature treatment, other mice were kept at a controlled temperature (23–25 °C) and humidity (40%), with 12-h day/night cycle, and fed a standard chow diet ad libitum. Only male mice were used for experiments and their littermate controls were used.

### Fluctuated temperature treatment
To perform the intervention of changed temperature, 12-month-old mice were randomly allocated into three groups, and exposed to room temperature (RT, 25 °C), low temperature (LT, 10 °C), and fluctuated temperature (FT) ranging from 10 to 25 °C. In detail, mice with FT treatment were put in a climatic chamber everyday, with automatic temperature control (25 °C at 8:00 am and 10 °C at 8:00 pm) and humidity (40%). No signs of stress or suffering were detected in any of the mice. After 12-week exposure, all mice were evaluated with exercise capacity tests, and sacrificed when the tests were finished.

### Feces microbiota transplantation (FMT)
Before FMT, 12-month-old mice were treated with antibiotic cocktail for 1 weeks. Antibiotic cocktail was composed of 100 μg/ml neomycin, 50 μg/ml streptomycin, 100 U/ml penicillin, 50 μg/ml vancomycin, 100 μg/ml metronidazole, 100 μg/ml CEFAZ, 125 μg/ml Ciprofloxine hydrochloride, 1 mg/ml bacitracin. For the microbiota suspension preparation, 5–6 fresh feces pellets from RT, LT, and FT mice were collected and homogenized in 1.2 ml reduced PBS (PBS with 0.5 g/l cysteine and 0.2 g/l Na$_2$S). After centrifugation at $600 \times g$ for 1 minute, 100 μl supernatant was gavaged into recipient mice (FMTrt, FMTlt, and FMTft). Mice were given microbiota suspension three times a week for 8 weeks, then examined through exercise capacity tests, and sacrificed when the tests were finished.

### α-AA treatment
12-month-old mice were received α-AA (MedChemExpress, HY-113328) freshly dissolved in drinking water at concentration of 500 mg/kg/d each mouse or received drinking water alone for 4 weeks. Food and water were given to the mice in ad libitum. After 4-week treatment, all mice were evaluated with exercise capacity tests, and sacrificed when the tests were finished.

### *Eubacterium* culture and oral supplementation
*Eubacterium* was grown in Fluid Thioglycolate Medium (Hopebio) in anaerobic condition and stored in 25% glycerol at concentration of $1 \times 10^{10}$ cfu/ml at −80 °C until use. The glycerol stocks were diluted in anaerobic PBS to the concentration of $1 \times 10^{9}$ cfu/ml. 12-month-old mice were exposed to the fluctuated temperature for 8 weeks, then orally gavaged with 100 μl either *Eubacterium* suspension or anaerobic PBS twice a week for 4 weeks. After 4-week treatment, all mice were evaluated with exercise capacity tests, and sacrificed when the tests were finished.

### Skeletal muscle function tests
(1) Grip test. Mice were placed on the center of wire grid, and then inverted 30 cm over a padded surface. Muscle strength was determined by the time spent hanging on the grid. Two repeats were performed on each mouse, and the maximum data were used.
(2) Rotarod test. Mice were acclimated to the rotarod with 6 rpm for 1 minute. Rotarod test was started from 6 rpm, and accelerated by 2 rpm per minutes until 10 minutes. The time mice spent on rotarod was recorded. Two repeats were performed on each mouse, and the maximum data were used.
(3) Treadmill. Mice were acclimated to the treadmill instrument at 10 m/s for 5 minutes in consecutive 3 days. Treadmill test was started from 10 m/s, and accelerated by 2 rpm per minutes until 24 m/s. The time when mice retired (when mice were unable to run or stayed on electric grid over 5 seconds) was recorded.

### Cell culture
C2C12 myoblasts (CL-0044, Procell) were cultured in Dulbecco's modified Eagle's medium (DMEM, Gibco) including glucose 25 mM, 10% fetal bovine serum (FBS, Gibico), 100 U/ml of penicillin and 100 μg/ml of streptomycin (Procell). Differentiation was induced by changing the 10% FBS to 2% horse serum for 2 days. All cells were cultured at 37 °C under a 5% CO$_2$ humidified atmosphere.

### Measurement of ROS level, ATP content, and mitochondrial membrane potential (MMP)
ROS were measured by using fluorescent probe DCFH-DA (Beyotime, S0033S). In vitro, DCFH-DA was loaded into C2C12 myotubes. After 37 °C incubation for 30 minutes, green fluorescence was examined using fluorescence microscope (Zeiss). In vivo, approximately 20 mg gastrocnemius was dissected, cut into pieces and homogenized in 2 mg/ml type I collagenase, followed by 60-minute digestion. After ending digestion through adding equal volume culture media with 2% FBS, the mixture was then filtered with 100-μm cell strainers and centrifuged at $200 \times g$ for 5 minutes. The precipitates were isolated and loaded with DCFH-DA, and the fluorescence was detected using Microplate Reader (Envision).

ATP contents were measured using ATP Assay Kit (Beyotime, S0027), following manufacturer's instructions. Briefly, cell suspension ($1 \times 10^{6}$) and gastrocnemius tissue (20 mg) were lysed with lysis buffer. The lysates were further centrifuged at $12,000 \times g$ for 5 minutes at 4 °C, and the supernatant was extracted for detection. The luminescence was further detected using Microplate Reader (Envision).

Mitochondrial membrane potential ($\Delta\psi_M$) was detected by JC-1 probe (Beyotime, C2006) and TMRE probe (Beyotime, C2001S). JC-1 was formed into J-aggregates (red fluorescence) in mitochondrial matrix under normal conditions, and existed as monomer (green fluorescence) when $\Delta\psi_M$ decreased. In vitro test, after 24-hour α-AA treatment, C2C12 myotubes were equipped with JC-1 probe, and incubated at 37 °C for 30 minutes. JC-1 fluorescence was further detected using fluorescence microscope (Zeiss). $\Delta\psi_M$ were calculated by the ratio of red to green fluorescence intensity. In vivo test, muscle mitochondria

were isolated using mitochondrial extraction kit (Beyotime, C3606), under guidance of kit's instructions. In short, 100 mg gastrocnemius were dissected, cut into pieces, and digested in 0.25% Tyrisin for 20 minutes. After $600 \times g$ centrifugation for 30 s, the precipitates were further homogenized in mitochondrial isolation buffer for 10–20 times. All the procedure above were conducted on ice. The homogenate was centrifuged at $600 \times g$ for 30 seconds, and the supernatants were isolated for further $11,000 \times g$ centrifugation. The precipitates were then isolated mitochondria, and 100 mg mitochondria were incubated with JC-1 probe. The JC-1 fluorescence was further tested by Microplate Reader (Envision). Similar procedures were also performed when using TMRE probe to investigate $\Delta\psi_M$.

### RNA extraction, reverse transcription, and real-time qPCR
Total RNA of cells and muscle tissue were prepared using TRIzol (Invitrogen). cDNA was prepared using Reverse Transcription Kit (Takara) following the manufacturer's instructions and RT-qPCR were performed using ABI QuantStudio3. For analysis of mtDNA/nuDNA ratio, mitochondrial genes (mt-Rnr2 and mt-Co2) and nuclear genes (Ucp2 and Hk2) were used. Primers were synthesized from Tsingke Biotech Co. Primers related with muscle atrophy, autophagy, mitophagy, mtDNA, and nuDNA were listed in Supplementary Table 2.

### Western blotting
Muscle tissue, cells, and mitochondria were lysed in RIPA Lysis Buffer (Biosharp), loaded on a 10–15% SDS-PAGE gel, and transferred to PVDF membrane (Millipore). Membranes were blotted with targeted primary antibodies as follows: anti-P62 (CST, 5114, 1:2000), anti-LC3B (CST, 2775, 1:1000), anti-PARK2 (Proteintech, 14060-1-AP, 1:500), anti-Ubiquitin (CST, 3933, 1:1000), anti-TFAM (Santa Cruz, sc-166965, 1:1000), anti-PGC1a (Santa Cruz, sc-518025, 1:1000), anti-MFN1 (Santa Cruz, sc-166644, 1:1000), anti-OPA1 (Santa Cruz, sc-393296, 1:1000), anti-phospho-mTOR (CST, 5536, 1:500), anti-mTOR (CST, 2983, 1:500), anti-TNFα (Santa Cruz, sc-52746, 1:1000), anti-IL6 (Santa Cruz, sc-28343, 1:1000), anti-GAPDH (OriGene, TA802519, 1:5000). After incubation overnight, secondary antibodies (Invitrogen, 31430 and 31460, 1:2000) were used. Immunoreactive proteins were detected using the Western ECL Substrate (BioRad) and quantified using Image Lab software (BioRad).

### HE, SDH, and BODIPY staining
For HE staining, GA muscles were fixed with paraformaldehyde (4%), dehydrated, and paraffin-embedded. Cross-sectional 5 μm thick slices of muscle were prepared and stained with hematoxylin and eosin (Servicebio) to observe the morphological changes. And cross-sectional areas were analyzed using Image J processing software.

For SDH and BODIPY staining, fresh GA muscles were embedded in OCT after isolation, then snap-frozen by isopentane with liquid nitrogen. 8-μm thick slices were prepared and stained with SDH solution (Solarbio) or BODIPY dye (3.8 μM working solution, Invitrogen).

### Autophagy flux detection in vivo and in vitro
In vitro, C2C12 myotubes were transfected with GFP-LC3 adenovirus, and subsequently treated with either 10 μM chloroquine (CQ, Sigma-Aldrich) or DMSO for 6 hrs. 48 hours later, all cells were stained with 200 nM MitoTracker (Beyotime, C1049B) for 20 minutes, and washed for three times. C2C12 myotubes were visualized using a confocal laser scanning microscope (Zeiss), and the overlapping (yellow) area was calculated. Three duplicates were conducted, and three images were taken per well.

In vivo, mice with α-AA treatment or its control were intraperitoneally injected with CQ at the dose of 60 mg/kg for 3 days, or PBS as the negative control. Three days later, muscle tissues were dissected and detected with autophagy markers via Western blot.

### Transmission electron microscopy (TEM)
Mitochondrial morphology in gastrocnemius muscle was analyzed by TEM. Gastrocnemius were cut into 1.0 mm³ pieces and fixed in 2.5% glutaraldehyde (servicebio) overnight. After rinsing three times, tissues were then incubated with 1% osmium tetroxide for 1 h. Tissues were then dehydrated in increasing concentrations of ethanol, embedded in epoxy. Embedded tissue was cut into ultra-thin sections (60-80 nm), which were then stained with 2% uranyl acetate and lead citrate and observed using a transmission electron microscope (Hitachi).

### Blood glucose, serum insulin, glucose tolerance test (GTT), insulin tolerance test (ITT), and homeostasis model assessment of insulin resistance index (HOMA-IR)
Blood glucose was measured by a glucometer monitor (Sinocare) and serum insulin level was detected by an ELISA kit (CUSABIO, CSB-E05071m). GTT and ITT were performed by intraperitoneal injection of 1 g/kg glucose after overnight fasting and 1 U/kg insulin after 4-h fasting respectively. Blood glucose levels were examined at 0 min, 15 min, 30 min, 60 min, 90 min after injection. Homeostasis model assessment of insulin resistance (HOMA-IR) index was calculated as previously reported[44].

### Metabolic parameter measurements and ambulation detection
Energy expenditure by indirect calorimetry and ambulation (activity) were evaluated using the automated comprehensive laboratory animal-monitoring system (Columbus Instruments) as previously reported[45]. Rectal temperature of mice was measured at 14:00 by a rectal probe attached to a digital thermometer (Physitemp, Clifton, NJ).

### Serum cortisol, adrenaline, and vasopressin measurements
Serum cortisol, adrenaline, and vasopressin levels were measured by cortisol ELISA kit (CUSABIO, CSB-E05113m), adrenaline ELISA kit (CUSABIO, CSB-E08679m), and vasopressin ELISA kit (CUSABIO, CSB-E09272m) respectively, according to the manufacturer's instructions.

### 16 S rDNA sequencing
4–5 feces pellets were obtained from each mouse of RT, LT, and FT. And the total genomic DNA was then extracted using DNA Extraction Kit following the instructions. The genome DNA was used as a template for PCR amplification of the V3-V4 hypervariable regions of the bacterial 16 S rRNA gene with the barcoded primers and Tks Gflex DNA Polymerase (Takara). Gel electrophoresis was then performed to detect amplicon quality. PCR products were purified with AMPure XP beads (Beckman), and further adjusted for sequencing. Sequencing was performed on an Illumina NovaSeq6000 with two paired-end read cycles of 250 bases each. (Illumina Inc.).

Clean reads were subjected to primer sequences removal and clustering to generate operational taxonomic units (OTUs) using Vsearch software with 97% similarity cutoff. The representative read of each OTU was selected using QIIME package. All representative reads were annotated and blasted against Silva database (Version 132) using RDP classifier (confidence threshold was 70%).

### Metagenome sequencing
Metagenomic libraries were prepared from 100 ngDNA using TruSeq Nano DNA Library Prep Kit (Illumina) and size selected at -350 bp. The pooled indexed library was sequenced using Illumina MiSeq. The raw data was in FASTQ format. Reads were trimmed and filtered using Trimmomatic(v0.36). Metagenome assembly was performed using MEGAHIT (v1.1.2) after getting valid reads. Use gaps inside scaffold as breakpoint to interrupt the scaffold into new contigs (Scaftig), and these new Scaftig with length ≥200 bp (or 500 bp) of were retained.

ORF prediction of assembled scaffolds using prodigal (v2.6.3) was performed and translated into amino acid sequences. The non-redundant gene sets were built for all predicted genes using CDHIT (v4.6.7). The clustering parameters were 95% identity and 90% coverage. The longest gene was selected as representative sequence of each gene set. Clean reads of each sample were aligned against the non-redundant gene set (95% identity) use bowtie2 (v2.2.9), and the abundant information of the gene in the corresponding sample was counted.

The gene set representative sequence (amino acid sequence) was annotated with NR, KEGG, COG, SWISSPROT, GO database with an e-value of 1e-5. The taxonomy of the species was obtained as a result of the corresponding taxonomy database of the NR Library, and the abundance of the species was calculated using the corresponding abundance of the genes. In order to construct the abundance profile on the corresponding taxonomy level, abundance statistics were performed at each level of Domain, Kingdom, Phylum, Class, Order, Family, Genus, Species. The 16 S rDNA sequencing and metagenome Sequencing, together with the corresponding analysis were conducted by OE Biotech Co., Ltd. (Shanghai, China).

### Statistical analysis

For comparison between two groups, Student's *t* test was used. Comparisons between multiple groups with one variable were calculated by one- way ANOVA with Tukey's test. Comparisons between multiple groups with not normal distribution were calculated by ANOVA non- parametric test (Kruskal–Wallis test and Dunn's Multiple Comparison post-test). *p* values assumed two-tailed distribution and unequal variances (*$P < 0.05$; **$P < 0.01$; ***$P < 0.001$). Statistical information relevant to individual experiments is detailed in the figure legends. GraphPad Prism 7 software was used for statistical analysis. The investigators were not blinded during group allocation and the experiment.

### Reporting summary

Further information on research design is available in the Nature Portfolio Reporting Summary linked to this article.

## Data availability

Data from CHARLS database can be obtained from official website (http://charls.pku.edu.cn/en). The 16 S rDNA gene sequences and metagenomic sequences were provided and available at National Center for Biotechnology Information Sequence Read Archive (SRA) database with accession code PRJNA962795 and PRJNA963010. Source data are provided in Source Data File with this paper.

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

## Acknowledgements

The authors would like to thank Professor Xianghang Luo of Xiangya Hospital, Central South University, for his good suggestions and critical reading. This work was supported by the National Natural Science Foundation of China (Grant Nos. 82000811 (Y.H.)). The geographical maps were created using ArcGIS online. And schematic diagram was conducted using BioRender.

## Author contributions

Y.H. designed the experiments, supervised the experiments, analyzed results, and co-wrote the manuscript; Y.L. carried out most of the experiments and drafted the manuscript; Y.G. helped to conduct animal and cell experiments; Z.L., R.Z., Y.H. helped to analyze CHARLS database and meteorological data; X.F., H.P., H.Z. proofread the manuscript, helped to generate and analyze data.

## Competing interests

The authors declare no competing interests.
