## [Peer Review File · Nature Communications]

Augmented temperature fluctuation aggravates muscular atrophy through the gut microbiotaREVIEWER COMMENTS

Reviewer #1 (Remarks to the Author):

In their manuscript, Liu et al. address whether temperature fluctuation affects the development of sarcopenia. The authors initially studied daily temperature fluctuation (DTF) and prevalence of sarcopenia in different regions of China and they found a positive correlation, with increase prevalence of sarcopenia in those regions with higher temperature fluctuation. These findings were the rationale to study the effects of DTF in the development of sarcopenia in mice. For that, they used middle-aged mice subjected to DTF (10-25°C) and analyzed different parameters associated with sarcopenia. They found that DTF increased muscle atrophy and worsened muscle function, and this was associated with changes in microbiota composition. Experiments with fecal transplantation showed that DTF-mouse microbiota aggravated sarcopenia in WT mice, which seems to be mediated by increased amino adipic acid. Finally, supplementation of eubacterium (a type of bacteria decreased by DTF) in DTF mice reversed the detrimental effects induced by DTF. Mechanistically, the authors propose that DTF aggravates sarcopenia by impairing mitophagy and inducing mitochondrial dysfunction. Although the role of microbiota during aging sarcopenia has already been studied, this study addresses an interesting topic, linking temperature effects with changes in microbiota and development of sarcopenia. However, there are several weaknesses the authors should address in order to convincingly demonstrate their conclusions.

Major points:

1. To address the role of temperature fluctuation, the authors used mice exposed to fluctuated temperature (FT, 10-25°C). To discard the effects caused by temperature per se, they used as a control, mice exposed to room temperature (RT, 25 °C) and low temperature (LT, 10 °C). An important issue in all the experiments performed with these groups is the body temperature of mice in each group. It is known that body temperature affects metabolic rate and longevity in mice, and therefore, it is an important parameter that should be considered. In addition, changes in metabolic homeostasis could also impact on muscle fitness. It would be important to monitor these parameters (body temperature, basal metabolic rate) in the three groups and evaluate whether they could have a role in sarcopenia.
2. When analyzing sarcopenia in FT mice (Fig 2), authors should also provide data on other molecular aspects that could be involved in its development: autophagy markers or even better autophagic flux, inflammatory markers, protein synthesis markers (mTOR, S6K) and protein degradation (proteasome). Those of them affected by FT, should be also analyzed in Figs 4, 6 and 8. Also, provide mean CSA values in all cases (a part from fiber distribution).
3. In Fig. 2h, if the bar color follows the same pattern than in the rest of the figure, the group with biggest changes in fiber size distribution is the LT group. This should be discussed. Moreover, no data on atrogenes expression is shown in Fig. 2k for this LT group.
4. In microbiota transplantation experiments, given that LT group also have important changes in microbiota compared with RT group, and it seems to have an effect on muscle fiber area (Fig. 2h), the authors should perform the same experiment with microbiota transplantation from LT mice.
5. One of the molecular mechanisms proposed by the authors by which FT increases sarcopenia is mitochondrial dysfunction. Further experiments are needed to support this notion. For example, authors evaluate ROS levels but in satellite cells, but this should be measured in muscle fibers. Other markers of mitochondrial mass should be considered, not only mtDNA. For mitochondrial membrane potential, they use JC-1 probe, and it is known that this probe leads to artifacts. It would be better to use TMRE for that purpose. In addition, mitochondrial respiration in permeabilized muscle fibers should be evaluated.
6. Regarding autophagy and mitophagy, the authors measured gene expression of some genes. This is not sufficient to evaluate these parameters. They should measure protein expression of autophagy and mitophagy markers and perform flux analyses by using lysosomal inhibitors (colchicine or chloroquine). In addition, mitochondrial dynamics is evaluated at gene expression level. Protein expression should be measured.

7. The effect of amino adipic acid on autophagy and mitophagy should be evaluated in vivo in mice by performing flux analyses as suggested above. In addition, flux analyses in cell culture experiments would help to show whether mitophagy/autophagy is inhibited.

Minor points:

1. Panels n and o in Fig.6 are explained in the text but are missing in the figure.
2. Line 232-233: "The mRNA levels of Atrogin-1, MurF1, Musa1 and myostatin were decreased after AA administration". The figure shows the opposite. Please correct.
3. Fig. 7: information about the color code for some panels is missing.
4. Some references are missing. For instance, information provided in lines 272-277 about the presence of eubacterium in elders and correlation with AA needs to be supported by references.
5. Discussion in general should be written in a more comprehensive way. Sometimes it is difficult to understand.
6. Discussion, line 291: the authors mention intermittent fasting, when they have not assess that in their manuscript.
7. Discussion, line 322: the authors state that AA treatment inhibited mitophagy in vivo and in vitro, when they have not perform any assay to show that in vivo.
8. Material and Methods section should be written in more detail. For example: at what time were mice sacrificed and samples obtained? How mitochondrial membrane potential has been measured in muscle? How mitochondrial fractions have been obtained?

Reviewer #2 (Remarks to the Author):

Reviewer Comments:

General Comments:

The reviewer thinks that this original article entitled "Augmented temperature fluctuation aggravates muscular atrophy through the gut microbiota" is of interest. The authors nicely resume their main scientific discoveries on how fluctuation of temperature can control muscle function through consecutively controlling mitophagy, bacterial metabolites secretion and bacterial abundancies. Hence their introduction and even their discussion are well structure. However, there are few limitations in the analysis in order to demonstrate the causality which link all this observation together. For example, the role of stress hormone in the model of augmented temperature fluctuation. Does the bacteria profile observed can be obtain with all the different type of stress coming from the environment (exercise, nutrition...)? The range of fluctuation is not also investigated in the preclinical model. Can you justified why you choose a DTR of 15 degrees? and not test 5, vs 10 vs 15 for example and why you choose 10-25 fluctuation? And note 20-35? Hence, the importance of mean of DTR need to be also discuss.

- In the clinical observation can you also give the absolute degree per geographical distribution. If you adjust the DTR by the mean temperature, does the difference remains?

If we follow your concept, you should see a different distribution of the population between continental vs oceanic climate. Can you check this information?

Do you know if in region with high DTR physical activity is lower?

- In typical analysis of cold exposure, OR temperature fluctuation, it is major to indicate the number of mice per cage because after acclimatization period, animal will regroup themselves into the cage to warm up and avoid shivering. Hence at the end, each mouse will not be really expose to 10C. Did you evaluate the time of regroupment or shivering? (Openfield record). You mention no signs of stress, however it would be pertinent to evaluate stress hormone (Cortisol, Adrenaline, ocytocine, Vasopressine) in the fluctuation group.

- A similar question related to physical activity quantification need to be address. After cold acclimatization, mice restart to move a little bit more, particularly with a 12 weeks protocol? Do you have record physical activity at different weeks baseline, 4, 8 12 weeks?

- Based on your short gun analysis you decide to focus on lysine degradation pathway. However you choice is not clearly explain particularly when we see all the other pathway affect in fig 6a. Can you justify your criteria of selection?

- In the same mine set, it will be useful to have in supplemental table the list of bacteria pointed out in your experiment and the one linked to α -AA and to the main path highlight in fig 6a. It will help to show which over bacteria could be responsible of the phenotype observed in your first set of experiment. Not convince that Eubacterium alone is doing all the job. It can also bring new information on the link between skeletal muscle specificity that your experiment also suggest. Example: Eubacterium seems to do not impact tibialis anterior (Fig 8) whereas you see an effect of the DTR in Figure 2. Can you discuss that part?

- Mitophagy is not a specific function of skeletal muscle. Did you have a look on the impact of Lysine degradation metabolite and amino adipic acid on mitochondria of other tissue such as the pancreas, liver or joint.

- One of another feature of sarcopenia is fat infiltration. In addition to the SDH staining can you add red oil staining? Particularly in the age group you tackle the 12-month-old mice.

- Based on the phenotype you show on muscle did you check the impact on glucose homeostasis?

Minor:

- Have look on typographical error through out the manuscript such in lines 271: "metagenomics"

- In different figure 2, 4, 6 and 8 when you talk about coordination, muscle endurance and treadmill, information need to be more develop in the legend. We do not understand muscle endurance assessment through grip strength ? In the panel Treadmill should be replace by maximal running time.....

- Figure 3. Number of mice per group in panel c seems only to be 4 in low temperature group. If you excluded several mice? you need to bring justification.

We would like to thank the reviewers for their thoughtful and constructive comments regarding our manuscript. We have addressed all of the questions and comments brought forth through additional experimentation and clarification. We hope that the reviewers will find our responses to their comments satisfactory, and we are willing to finish the revised version of the manuscript including any further suggestion that the reviewers may have.

The following responses have been prepared to address the two reviewers' comments in a point-by-point fashion.

Reviewer #1 (Remarks to the Author):

In their manuscript, Liu et al. address whether temperature fluctuation affects the development of sarcopenia. The authors initially studied daily temperature fluctuation (DTF) and prevalence of sarcopenia in different regions of China and they found a positive correlation, with increase prevalence of sarcopenia in those regions with higher temperature fluctuation. These findings were the rationale to study the effects of DTF in the development of sarcopenia in mice. For that, they used middle-aged mice subjected to DTF (10-25oC) and analyzed different parameters associated with sarcopenia. They found that DTF increased muscle atrophy and worsened muscle function, and this was associated with changes in microbiota composition. Experiments with fecal transplanted showed that DTF-mouse microbiota aggravated sarcopenia in WT mice, which seems to be mediated by increased amino adipic acid. Finally, supplementation of eubacterium (a type of bacteria decreased by DTF) in DTF mice reversed the detrimental effects induced by DTF. Mechanistically, the authors propose that DTF aggravates sarcopenia by impairing mitophagy and inducing mitochondrial dysfunction. Although the role of microbiota during aging sarcopenia has already been studied, this study addresses an interesting topic, linking temperature effects with changes in microbiota and development of sarcopenia. However, there are several weaknesses the authors should address in order to convincingly demonstrate their conclusions.

Major points:

1. To address the role of temperature fluctuation, the authors used mice exposed to fluctuated temperature (FT, 10-25oC). To discard the effects caused by temperature per se, they used as a control, mice exposed to room temperature (RT, 25 oC) and low temperature (LT, 10 oC). An important issue in all the experiments performed with these groups is the body temperature of mice in each group. It is known that body temperature affects metabolic rate and longevity in mice, and therefore, it is an important parameter that should be considered. In addition, changes in metabolic homeostasis could also impact on muscle fitness. It would be important to monitor these parameters (body temperature, basal metabolic rate) in the three groups and evaluate whether they could have a role in sarcopenia.

Response:

We appreciate Reviewer #1's constructive comments. As shown in the figure below, mice put in 10°C had decreased level of core temperature, compared to RT or

FT mice. Meanwhile, we used energy expenditure (EE) and O₂ consumption to reflect the level of resting metabolic rate (RMR), and found LT mice had increased energy expenditure. No significant difference was found in these parameters between RT and FT group, for which there were several possible explanations: 1) FT mice were put in room temperature half of the day, the level of EE might return to normal status during that stage; 2) FT treatment might enhance EE; however, other factors resisted the effects, such as enhanced mitophagy in metabolically crucial organs. Thus, we assumed that dampened muscle function caused by FT treatment were not mediated by altered altered metabolic level.

2. When analyzing sarcopenia in FT mice (Fig 2), authors should also provide data on other molecular aspects that could be involved in its development: autophagy markers or even better autophagic flux, inflammatory markers, protein synthesis markers (mTOR, S6K) and protein degradation (proteasome). Those of them affected by FT, should be also analyzed in Figs 4, 6 and 8. Also, provide mean CSA values in all cases (a part from fiber distribution).

Response:

We appreciate Reviewer #1's constructive comments.

1) We detected aforementioned markers via Western blot in each mouse model, and attached the results in each part. In FT mouse model, we found the decreased level of autophagy- and mitophagy-related markers (P62, LC3II/I and PARK2) (Figure 5i). And no significant difference was found in biomarker levels of protein synthesis (phospho/total-mTOR), protein degradation (Ubiquitin) and inflammation (TNF α , IL6). Decreased autophagy and mitophagy- related markers were also decreased in FMTft mice and α AA-treated mice, compared to their controls (Figure S3b-c). In *Eubacterium* mouse model, *Eubacterium* administration caused increased level of LC3II/I and PARK2 (Figure S3e). Other parameters revealed no significant difference.

Figure S3

2) We added mean CSA in Figure 2g, 4f, 6j, 8h, and relevant description in manuscript.

3. In Fig. 2h, if the bar color follows the same pattern than in the rest of the figure, the group with biggest changes in fiber size distribution is the LT group. This should be discussed. Moreover, no data on atrogenes expression is shown in Fig. 2k for this LT group.

Response:

We are deeply sorry for that we labeled a wrong bar. We have carefully checked and revised it in Figure 2i. Fiber size distribution revealed a significant decrease in the number of large diameter fibers (>3000 μm^2) and an increase in that of smaller fibers (500-1500 μm^2) in muscle of FT mice, instead of LT mice. And the expression of atrogenes were attached in Figure 2l.

Combined with decreased gastrocnemius mass and elevated *Atrogin1*, *MuRF1* expression in LT mice (Figure 2e, 2l), we assumed that LT treatment partly affected muscle function. However, FMT tests revealed that LT-shaped gut microbiota didn't mimic the effects of LT on muscle function (Figure 3, See the reply in Point 4), suggesting LT-related muscle change might not in gut microbiota- dependent way. Other factors might contribute to the change, such as decreased physical activity (S21).

4. In microbiota transplantation experiments, given that LT group also have important changes in microbiota compared with RT group, and it seems to have an effect on muscle fiber area (Fig. 2h), the authors should perform the same experiment with microbiota transplantation from LT mice.

Response:

We performed fecal microbiota transplantation (FMT) experiment from RT, LT, FT mice to three groups of recipient mice. As shown in Fig4, gut microbiota from LT group had little effects on muscle fiber area and function when compared with RT mice. However, FMTft mice had decreased muscle mass and exercise capacity, compared with either FMTrt or FMTlt mice.

5. One of the molecular mechanisms proposed by the authors by which FT increases sarcopenia is mitochondrial dysfunction. Further experiments are needed to support this notion. For example, authors evaluate ROS levels but in satellite cells, but this should be measured in muscle fibers. Other markers of mitochondrial mass should be considered, not only mtDNA. For mitochondrial membrane potential, they use JC-1 probe, and it is known that this probe leads to artifacts. It would be better to use TMRE for that purpose. In addition, mitochondrial respiration in permeabilized muscle fibers should be evaluated.

Response:

1) For ROS detection in vivo, we isolated single cells from muscle tissue through mechanical shearing and collagenase digestion, then loaded them with DCF probe. And DCF probe fluorescence were detected through microplate reader. In this case, the detected fluorescence was mainly reflected the ROS level of digested cells from muscle tissue. We found the increased ROS level in digested cells from FT and FMTft mice.

2) To evaluate mitochondrial mass, we further analyzed the fluorescence of Mitotracker of α -AA- or DMSO- treated C2C12 cells, and found the increased level of mitochondrial contents in cells exposed to 100 μ M α -AA.

3) We further used TMRE probe to analyze MMP, and found similar results conducted by JC-1 probe, that FT/FMTft mice had decreased level of MMP in isolated mitochondria, compared with RT/FMTTrt groups.

4) Unfortunately, we only conducted mitochondrial respiration in C2C12 based on the Seahorse XF96 Extracellular Flux we have. And we found that α -AA exposure resulted in reduced cellular respiration under both basal and uncoupled conditions compared with the controls.

6. Regarding autophagy and mitophagy, the authors measured gene expression of some

genes. This is not sufficient to evaluate these parameters. They should measure protein expression of autophagy and mitophagy markers and perform flux analyses by using lysosomal inhibitors (colchicine or chloroquine). In addition, mitochondrial dynamics is evaluated at gene expression level. Protein expression should be measured.

Response:

1) We added western blotting results of autophagy markers and mitochondrial dynamics markers in corresponding parts.

Line234-235: ‘Protein level of TFAM, PGC1 α , (biogenesis), and MFN1, OPA1 (dynamics) showed no significant difference between muscle of FT and RT mice (Figure 5i-j). Protein level related to autophagy and mitophagy, including P62, LC3 and PARK2 were decreased in FT mice (Figure 5i-j).’

2) We conducted in autophagy flux test in α -AA-treated mice and cells.

(In vivo) Line273-277: ‘The blockage of autophagic flux after α -AA treatment was showed using lysosomal inhibitor chloroquine (CQ), which led to reduced level of LC3-II (Figure S3d). And mice with both α -AA and CQ administration had lower level of autophagy markers compared with control group with CQ treatment alone (Figure S3d).’

(In vitro) Line297-301: ‘Furthermore, to inhibit autophagy flux, C2C12 cells treated with GFP-LC3 adenovirus and Mitotracker were exposed to CQ. Co-staining of red fluorescence (mitochondria) and green fluorescence (autophagosomes) were lower in cells with α -AA/CQ treatment, compared with cells with both DMSO/CQ treatment, indicating the decreased mitophagy flux led by α -AA supplement (Figure 7c-d).’

7. The effect of amino adipic acid on autophagy and mitophagy should be evaluated in vivo in mice by performing flux analyses as suggested above. In addition, flux analyses in cell culture experiments would help to show whether mitophagy/autophagy is inhibited.

Response:

We have conducted flux tests and presented the results in Point 6. The mitophagy flux revealed the decreased level of mitophagy led by α -AA.

Minor points:

1. Panels n and o in Fig.6 are explained in the text but are missing in the figure.

Response:

We apologize deeply for our mistake and added the corresponding results in Figure 6n-o.

Line 270-271 ‘Additionally, reduced $\Delta \psi M$ and ATP contents were also found in gastrocnemius of αAA -treated mice (Figure 6q-r)’

2. Line 232-233: “The mRNA levels of Atrogin-1, MurF1, Musa1 and myostatin were decreased after AA suministration”. The figure shows the opposite. Please correct.

Response:

We apologize deeply for our mistake and corrected it in revised manuscript (Line: 268-270 ‘The mRNA levels of *Atrogin1* and *Myostatin* were elevated after αAA administration (Figure 6p).’).

3. Fig. 7: information about the color code for some panels is missing.

Response:

We corrected it in revised figure.

4. Some references are missing. For instance, information provided in lines 272-277 about the presence of *Eubacterium* in elders and correlation with AA needs to be supported by references.

Response:

We added the missing reference and more information in that part.

Line319-328: ‘We searched the decreased bacteria in feces FT mice from gut metagenomic profiles, and chose *Eubacterium* for the further study, for the reason that 1) The level *Eubacterium* was negatively associated with the bacteria had high LDA score in FT-treated mice, including *Bacteroides*, *Paraprevotella* (Figure 2f, Figure S5a) ; 2) Decreased *Eubacterium* level in FT group were partly contributing to the Lysine degradation pathway based on metagenomic results (Figure S5b). Therefore, we further inquired whether *Eubacterium* supplement could improve the muscle capacity of mice exposed to fluctuated temperature.’

Line 375-377: ‘*Eubacterium* were reported to be enriched in individuals with exercise habits, decreased in elder people with physical frailty and sarcopenia⁴⁰⁻⁴².’

5. Discussion in general should be written in a more comprehensive way. Sometimes it is difficult to understand.

Response:

Thanks again for Reviewer #1’s suggestion, we have revised discussion part thoroughly.

6. Discussion, line 291: the authors mention intermittent fasting, when they have not assess that in their manuscript.

Response:

We apologize deeply for our mistake and delete it in revised manuscript.

7. Discussion, line 322: the authors state that AA treatment inhibited mitophagy in vivo and in vitro, when they have not perform any assay to show that in vivo.

Response:

Thanks again for Reviewer #1’s remind. We attached Western blot result of mitophagy markers and autophagy flux in Figure S3. In Figure S3c, there were decreased levels of p62, LC3II/I and PARK2 in FT mice. In Figure S3d, autophagy flux results showed that mice with both α -AA and CQ administration had lower level of autophagy markers compared with control group with CQ treatment alone. These findings indicate the deceased mitophagy level led by α -AA administration in vivo.

8. Material and Methods section should be written in more detail. For example: at what time were mice sacrificed and samples obtained? How mitochondrial membrane potential has been measured in muscle? How mitochondrial fractions have been obtained?

Response:

Thanks for Reviewer #1's suggestions, and we add more details in Method. We added sacrifice time in Line 447, 460, 467 and 476, and elucidated the details of MMP measurements and mitochondrial fractions extraction in Line 519-536: 'Mitochondrial membrane potential ($\Delta\psi_M$) were detected by JC-1 probe (Beyotime, C2006) and TMRE probe (Beyotime, C2001S). JC-1 were formed into J-aggregates (red fluorescence) in mitochondrial matrix under normal conditions, and existed as monomer (green fluorescence) when $\Delta\psi_M$ decreased. In vitro test, after 24-hour α -AA treatment, C2C12 myoblast were equipped with JC-1 probe, and incubated in 37 °C for 30 minutes. JC-1 fluorescence were further detected using fluorescence microscope (Zeiss). $\Delta\psi_M$ were calculated by the ratio of red to green fluorescence intensity. In vivo test, muscle mitochondria were isolated using mitochondrial extraction kit (Beyotime, C3606), under guidance of kit's instructions. In short, 100mg gastrocnemius were dissected, cut into piece, and digested in 0.25% Tyrisin for 20 minutes. After 600g centrifugation for 30 seconds, the precipitates were further homogenized in mitochondrial isolation buffer for 10-20 times. All the procedure above were conducted on ice. The homogenate were centrifuged at 600g for 30 seconds, and the supernatants were isolated for further 11000g centrifugation. The precipitates were the isolated mitochondria, and 100mg mitochondria were incubated with JC-1 probe. The JC-1 fluorescence were further tested by Microplate Reader (Envision). Similar procedure were also performed when using TMRE probe to investigate $\Delta\psi_M$.'

Reviewer #2 (Remarks to the Author):

Reviewer Comments:

The reviewer thinks that this original article entitled “Augmented temperature fluctuation aggravates muscular atrophy through the gut microbiota” is of interest. The authors nicely resume their main scientific discoveries on how fluctuation of temperature can control muscle function through consecutively controlling mitophagy, bacterial metabolites secretion and bacterial abundancies. Hence their introduction and even their discussion are well structure. However, there are few limitations in the analysis in order to demonstrate the causality which link all this observation together. For example, the role of stress hormone in the model of augmented temperature fluctuation. Does the bacteria profile observed can be obtain with all the different type of stress coming from the environment (exercise, nutrition...) ?

Response:

We appreciate Reviewer #1’s the constructive comments.

1) We detected circulating stress hormone, including Cortisol, Adrenaline and Vasopressin, and found no significant difference among these groups after 12-week treatment. We assumed that there might be a transient alteration in stress hormone level in response to the changed ambient temperature, and the alteration gradually diminished after adaption to the temperature¹⁻².

2) On the one hand, different stress can lead to the unique bacteria profile, for instance, LT treatment led to increased *Ruminococcaceae* and decreased *Muribaculaceae* in family level, which were consistent with published literature conducted in cold environment. On the other hand, FT mice had similar level of food intake (Figure S2a) and physical activity (Figure S2l), compared with RT mice, indicating the altered profile not resulted from changed nutrition or exercise. Based on the characteristic bacteria composition in FT mice, we tended to think the main effective factor were the changed temperature.

The range of fluctuation is not also investigated in the preclinical model. Can you justified why you choose a DTR of 15 degrees ? and not test 5, vs 10 vs 15 for example and why you choose 10-25 fluctuation? And note 20-35 ? Hence, the importance of mean of DTR need to be also discuss.

Response:

We added corresponding parts in discussion section Line 349-356: ‘Increased DTR was reported to be a risk factor of many diseases based on epidemiological data. Tang et al. reported the significantly positive relationship between long-term DTR exposure and all-cause mortality, with a 13% higher risk of all-cause mortality based on a 1 °C increase in the DTR exposure. In our study, we chose 10-25°C for the mouse model, based on 1) the maximum mean DTR in all the cities included were 15.55 °C; 2) the mean daily maximum temperature is 21.6 °C, and the minimum is 11.9 °C. The set of 25 °C in day time and 10 °C in night time were more appropriate to mimic the effects of daily temperature change on Chinese.’

- In the clinical observation can you also give the absolute degree per geographical distribution. If you adjust the DTR by the mean temperature, does the difference remains?

Response:

1) We attached the absolute degree of mean DTR in Supplementary Table 2 as follows:

Supplementary Table 2
Mean DTR in 2011-2013

Province	City	Mean DTR (°C)	Province	City	Mean DTR (°C)
Anhui	Anqing	8.32	Jiangsu	Yancheng	8.59
	Bozhou	9.60		Yangzhou	8.88
	Fuyang	10.09	Jiangxi	Nanchang	7.35
	Huainan	8.35		Ganzhou	8.44
	Liuan	9.18		Jian	8.49
	Suzhou	9.42		Jingdezhen	9.26
	Beijing	9.88		Jiujiang	7.65
Chongqing	Chongqing	7.26	Shangrao	9.42	
Fujian	Fuzhou	7.78		Yichun	8.68
	Ningde	7.17	Liaoning	Anshan	8.26
	Putian	6.78		Benxi	10.25
	Zhangzhou	8.06		Chaoyang	12.43
Gansu	Lanzhou	13.27		Dalian	6.25
	Dingxi	12.50		Jinzhou	9.65
	Pingliang	11.89	Neimenggu	Hohhot	11.93

	Zhangye	15.55		Chifeng	11.88
Guangdong	Guangzhou	7.79		Hulunbuir	13.26
	Chaozhou	8.47		Hinggan	10.88
	Maoming	7.56		Xilingol	12.91
	Qingyuan	7.16	Qinghai	Haidong	14.24
	Jiangmen	6.57		Jinan	9.11
	Shenzhen	6.02		Binzhou	10.65
	Foshan	6.43	Shandong	Dezhou	9.66
Guangxi	Nanning	7.88		Liaocheng	10.50
	Yulin	7.47		Linyi	10.00
	Guilin	7.19		Qingdao	5.99
	Hechi	7.74		Weihai	6.56
Guizhou	Qiannan	7.84		Weifang	10.41
	Qiandongnan	9.06		Zaozhuang	9.62
Hebei	Shijiazhuang	8.78	Shanxi	Linfen	11.45
	Baoding	10.91		Xinzhou	13.70
	Cangzhou	10.25		Yangquan	12.00
	Chengde	13.25		Yuncheng	11.74
Henan	Zhengzhou	9.98	Shanghai	Shanghai	7.39
	Anyang	10.77	Shaanxi	Baoji	10.36
	Jiaozuo	10.31		Hanzhong	8.79
	Luoyang	9.88		Weinan	10.77
	Pingdingshan	10.18		Yulin	11.86
	Puyang	10.59	Sichuan	Chengdu	8.78
	Xinyang	8.86		Ganzi	15.01
	Zhoukou	9.47		Guangan	7.39
Heilongjiang	Harbin	9.75		Nanchong	7.68
	Jixi	9.73		Liangshan	11.90
	Jiamusi	10.48		Mianyang	7.90
	Qiqihar	10.12		Neijiang	7.78
Hubei	Enshi	8.32		Yibin	7.49
	Huanggang	8.16		Ziyang	7.86
	Jingmen	8.18		Meishan	7.91
	Xiangfan	8.80	Tianjin	Tianjin	9.82
Hunan	Changsha	7.42	Xinjiang	Akesu	12.90
	Changde	8.56	Yunnan	Kunming	11.05
	Loudi	8.51		Baoshan	11.63
	Shaoyang	8.45		Chuxiong	11.01

	Yiyang	7.94		Lijiang	12.24
	Yueyang	6.45		Lincang	12.28
Jilin	Jilin	9.27		Zhaotong	9.93
	Siping	9.93	Zhejiang	Hangzhou	8.42
Jiangsu	Lianyungang	9.33		Huzhou	8.84
	Suzhou	7.91		Jiaxing	8.13
	Suqian	8.87		Lishui	10.21
	Taizhou	8.65		Ningbo	8.56
	Xuzhou	9.54		Taizhou	7.56

2) We conducted partial correlation analysis by using mean temperature of year 2011-2013 in aforementioned cities as adjustment, and added the relevant details.

Line 105-108: 'To eliminate the effects of local temperature *per se*, we used local mean temperature as adjustment, and discovered the positive correlation among possible sarcopenia prevalence and mDTR still remained (Total: $R=0.33$ $P=2.01E-04$, Male: $R=0.237$ $P=8.44E-03$, Female: $R=0.314$ $P=4.20E-04$).'

If we follow your concept, you should see a different distribution of the population between continental vs oceanic climate. Can you check this information?

Response:

Unfortunately, we didn't find the published data on sarcopenia prevalence conducted in both continental vs oceanic climate countries. Thus, we searched the epidemiological data and sarcopenia prevalence in representative cities (listed below). However, on account of the differences in diagnostic criteria, age and race *etc.* among these studies, it is inappropriate to compare the prevalence of sarcopenia in cities with different climates directly. Meta-analysis or cross-sectional studies based on same criteria are needed to elucidate the issue in the future.

Country	Year	Mean Age	Male prevalence	Female prevalence	Total prevalence	Diagnostic criterion	PMID
Maritime Climate							
The UK	2019	65	208/1564 (13.3%)	287/1840 (15.6%)	495/3404 (14.5%)	Grip Strength (Women < 16 kg, Men < 26 kg)	30368480
Germany	2013	68.1±3.6	160/622 (25.7%)	181/783 (23.1%)	341/1405 (24.3%)	SMI (Men≤7.26 kg/m ² , Women≤5.5 kg/m ²)	25877773
Switzerland	2019	83.6 (± 5.6)	23/82 (28%)	36/137 (26.3%)	59/219 (26.4%)	handgrip strength (women< 16 kg, Men< 27 kg)	35794312
France	2017	80.5 ± 3.9		504/3025 (16.7%)		SARC-F Questionnaire	28629717
Ireland	2019	81.7	7/51 (13.7%)	28/82 (34.1%)	35/133 (26.3%)	①ASM(Women <6kg/m ² , Men < 7kg/m ²); ②SPPB <8 and/or TUG>20s	33583001
Netherlands	2019	61.8 ± 4.3	228/724 (31.5%)	360 /771 (46.7%)	588/1495 (39.3%)	①grip strength(men<30kg, women<20kg) ②Arm-to-leg BIA data(M<10.75 kg/m ² ,F:<6.75 kg/m ²)	32894680
Belgium	2014	73.5±6.16	25/212 (11.8%)	48/322 (14.9%)	73/534 (13.7%)	①SMI (men<7.26 kg/m ² , women<5.5 kg/m ²); ②grip strength(Men<30kg, Women<20kg); ③SPPB test≤8	25979160
Denmark	2016	75±7	8/323 (3.5%)	6/297 (2.1%)	14/529 (2.7%)	①HGS(women < 16 kg, Men < 27 kg); ②ALM/h ² (women ≤5 kg/m ² /Men ≤6.6 kg/m ²)	32464171
Poland	2020		109/823 (13.2%)	263/1177 (22.3%)	372/2000 (18.65%)	SARC-F score value ≥ 4	36079726
Sweden	2020	86	11/35 (31.4%)	16/57 (28.1%)	27/92 (29%)	①SARC-F score value ≥ 4; ②<10/8 chair stands in 30 sec (<85 yrs) or <8 chair stands in 30 sec (>85 yrs)	33331617
Austria	2018	80.6 ± 5.5	15/57 (26.3%)	4/84 (4.7%)	19/141 (13.5%)	①gait speed ≤0.8m/s; ② Grip Strength (Women < 20 kg, Men < 30 kg) ③ASM(men< 7.26 kg/m ² ,women <5.5 kg/m ²)	31049683
Chile	2008	69.2±6.9	134/756 (17.7%)	249/1555 (16%)	467/2311 (20.2%)	①SMI men<7.45 kg/m ² ; women<5.88 kg/m ² ; ②grip strength(Women<5 kg, Men<27 kg); ③gait speed ≤ 0.8m/s	33883888
New Zealand	2019	86.0 ± 8.3	14/33 (42.4%)	23/58 (39.7%)	37/91 (41%)	①Grip Strength (Women < 16 kg, Men < 27 kg) ; ②appendicular muscle mass/height ² (Men<7 kg/m ² , Women<5.5 kg/m ²)	35565805
Continental climate							
Iran Deheran	2011	66.8	32/150 (21.3%)	22/150 (14.7%)	54/300 (18%)	①ASM(Women<5.45 kg/m ² , Men<7.26 kg/m ²); ②hand grip strength	35383191
Hungary	2019	66		31/100 (31%)		①SARC-F score value ≥ 4,②Grip Strength,③ASM (<15kg) ,	35246081
Argentina		70.4 ± 7.7		10/250 (4%)		①ASM/height ² <5.5 kg/m ² or 15 kg; ②HGS< 16 kg ; ③gait speed ≤0.8m/s , sit-to-stand test > 15 seconds	33611244
Finland Helsinki	2018	87	27/126 (21.4%)			①Grip Strength < 27 kg ; ②appendicular muscle mass < 20 kg ; ③SPPB test≤8 points	32376097
Czech Republic	2012	83.0 ±6.3			15/77 (19.5%)	calf circumference <31 cm	27925146
Canada Guelph	2012	75.2 ± 5.7	2/42 (4.7%)	3/43 (7.1%)	5/85 (5.9%)	①gait speed <0.8 m/s ; ②grip strength(Men<30kg, Women<20kg)	23322451
Russia	2010	72.2			135/1663 (8.1%)	①HGS(women < 16 kg/Men < 27 kg); ②SMI	34219725

Do you know if in region with high DTR physical activity is lower ?

Response:

To clarify the relationship between DTR and physical activity, we used the data from CHARLS 2013, which included questionnaire to measure participants' physical activity (PA) levels based on International Physical Activity Questionnaire (IPAQ), and conducted correlation analysis.

Briefly, participants were asked whether they had performed at least 10 min of vigorous physical activity (VPA, such as carrying heavy loads, digging, plowing, aerobic exercise, fast cycling, bicycling with cargo, etc.), moderate physical activity (MPA, such as carrying light things, cycling at regular speed, mopping, tai chi, brisk walking, etc.), and low-intensity physical activity (LPA, such as walking at work or at

home and walking for recreation, exercise, or leisure), and their weekly frequency, time spent on physical activity.

The total physical activity volume (PAV) score can be expressed using the metabolic equivalent (MET), which is calculated as follows: $PAV = 8.0 \times \text{weekly VPA duration score} + 4.0 \times \text{weekly MPA duration score} + 3.3 \times \text{weekly LPA duration score}$. The weekly duration scores were the total time spent on the PAs per week. We transformed the time range into the middle value as follows, “ ≥ 10 min and < 30 min” was recorded as 20 min, “ ≥ 30 min and < 2 h” was recorded as 75 min, “ ≥ 2 h and < 4 h” was recorded as 180 min, and “ ≥ 4 h” was recorded as 240 min³⁻⁴.

According to the IPAQ, physical inactivity (PI) was indicated if the total PAV did not reach 600 MET-minutes/week. Thus, we calculated local PI% (participants with PI/total participants) in aforementioned cities. Scatterplot revealed that no significant correlation between PI% and mDTR, indicating that people lived in high DTR cities not necessarily had low physical activity.

- In typical analysis of cold exposure, OR temperature fluctuation, it is major to indicate the number of mice per cage because after acclimatization period, animal will regroup themselves into the cage to warm up and avoid shivering. Hence at the end, each mouse will not be really expose to 10°C. Did you evaluate the time of regroupment or shivering? (Openfield record). You mention no signs of stress, however it would be pertinent to evaluate stress hormone (Cortisol, Adrenaline, ocytocine, Vasopressine) in the fluctuation group.

Response:

1) We evaluated the time when mice regrouped together, and found mice under 10°C regrouped more often, compared with mice under room temperature. Regroupment affected actual temperature treated with mice. We searched the published literature which also put mice under 10°C (in separated cage), and found that the energy expenditure in mice under 10°C was approximately 1.5 times higher than mice in room temperature⁵⁻⁶, which were also shown in LT mice when compared with RT mice (Figure S2c) (The energy expenditure/metabolic rate of mice were sensitive to ambient temperature, reflecting the response to different temperature). Meanwhile, we mainly focused on the effects brought from temperature changing, instead of the temperature *per se*. Thus, we think the effects of 10°C with regroupment were similar with that of 10°C alone.

2) We detected the level of circulating stress hormone after 12-week treatment, including cortisol, adrenaline and vasopressin, and found no significant difference among these groups. We assumed that there might be a transient alteration in these hormone level in response to the changed ambient temperature, and the alteration gradually diminished after adaption to the temperature¹⁻².

- A similar question related to physical activity quantification need to be address. After cold acclimatization, mice restart to move a little bit more, particularly with a 12 weeks protocol? Do you have record physical activity at different weeks baseline, 4, 8 12 weeks?

Response:

We monitored their locomotion through Oxymax/CLAMS system after 12-week treatment. As shown below, LT mice had decreased physical activity after 4- and 8-week treatment compared with RT mice, while the levels of physical activity in 4th week, 8th week and 12th week were similar. No significant difference was found in the physical activity between FT and RT mice, suggesting that dampened muscle function not resulted from decreased physical activity.

- Based on your short gun analysis you decide to focus on lysine degradation pathway. However your choice is not clearly explain particularly when we see all the other pathway affect in fig 6a. Can you justify your criteria of selection?

Response:

Gut microbiota can exert its effects through metabolites, including SCFA, amino acids, etc. Here, among all the contributing KEGG pathway in FT groups, we pay attention to the lysine degradation, since lysine was closely related with the onset of sarcopenia⁷⁻⁹. Reduced lysine level was found in muscle of both aged and sedentary mouse model, and regarding as a therapeutic target in combating aging-related disease¹⁰. And lysine supplement was reported to enhance muscle strength¹¹⁻¹². Here, inspired by enriched lysine degradation pathway, we detected lysine level and found no difference between RT and FT mice. However, FT mice had elevated lysine degradation product, α -AA. Thus, we chose α -AA for further study.

- In the same mine set, it will be useful to have in supplemental table the list of bacteria pointed out in your experiment and the one linked to α -AA and to the main path highlight in fig 6a. It will help to show which over bacteria could be responsible of the phenotype observed in your first set of experiment. Not convince that Eubacterium alone is doing all the job. It can also bring new information on the link between skeletal muscle specificity that your experiment also suggest. Example: Eubacterium seems to do not impact tibialis anterior (Fig 8) whereas you see an effect of the DTR in Figure 2.

Can you discuss that part?

Response:

We searched the bacteria contributing to the KEGG pathway with high LDA score in FT group, and listed top 10 bacteria in Figure S5c. In family level, *Lachnospiraceae*, *Muribaculaceae*, etc were contributed to PI3K-AKT Signaling Pathway (the Top2 enriched KEGG pathway) and Insulin Signaling Pathway, which were also contributed to Lysine Degradation Pathway. Besides, we searched the decreased bacteria in FT group which enriched in Lysine Degradation Pathway, and found *Eubacterium* were one of them, indicating decreased *Eubacterium* can be related with altered Lysine Degradation Pathway.

And we added corresponding contents in Discussion part: ‘To overcome dampened

muscle function led by FT treatment, we selected the *Eubacterium*, a decreased bacteria in FT group. *Eubacterium* were reported to be enriched in individuals with exercise habits, decreased in elder people with physical frailty and sarcopenia⁴⁰⁻⁴². Here, based on the correlation analysis (Figure S5a), we found *Eubacterium* were negatively correlated with the bacteria had high LDA score in FT-treated mice, including *Bacteroides*, *Paraprevotella*, and positively correlated with the beneficial bacteria on muscle function, such as *Ruminococcus*⁴³. *Eubacterium* supplement significantly increased muscle mass, including quadriceps femoris and gastrocnemius (GA). The mass of tibialis anterior (TA) were not elevated in statistically different way, which suggested that reversing the decreased *Eubacterium* level alone can't entirely restore the muscle dysfunction caused by FT treatment. As listed in Figure S5c, the bacteria species belonging to *Lachnospiraceae*, *Muribaculaceae* also contributed to the altered KEGG pathway in FT group, indicating other bacteria might also participate in the FT-related sarcopenia.'

- Mitophagy is not a specific function of skeletal muscle. Did you have a look on the impact of Lysine degradation metabolite and amino adipic acid on mitochondria of other tissue such as the pancreas, liver or joint.

Response:

We evaluated the effects of α -AA treatment on mitochondria of liver, since its well-functioning (e.g:free fatty acid β -oxidation) rely on mitochondria. We detected

MMP and ATP levels in liver of α -AA-treated mice, and found the decreased hepatic TMRE and ATP levels as well, suggesting that α -AA treatment not only acted on muscle tissue. α -AA supplement were reported to impair insulin signaling in both cell and mouse models¹³, however, other study showed the opposite results¹⁴. Thus, the effects of α -AA might be diverse, and still need further exploration.

- One of another feature of sarcopenia is fat infiltration. In addition to the SDH staining can you add red oil staining? Particularly in the age group you tackle the 12-month-old mice.

Response:

We evaluated fat infiltration through BODIPY staining, and added the results in Line 266-268: ‘Since fat infiltration was also a typical characteristic of sarcopenia, we stained muscle tissue with BODIPY dye, and noticed the increase of stained lipid droplet in AA-treated muscle (Figure 6n-o).’

- Based on the phenotype you show on muscle did you check the impact on glucose homeostasis?

Response:

GTTs and ITTs revealed the reduced blood glucose level of LT mice, indicating the better insulin sensitivity and glucose tolerance led by cold stress (10°C). Considered LT mice also had relatively decreased HOMA-IR index (though not in significant way), we inferred LT mice had better performance in glucose homeostasis. For FT mice, no significant difference was found in these tests compared with RT mice, indicating fluctuated temperature didn’t have significant impact on glucose homeostasis.

Minor:

- Have look on typographical error through out the manuscript such in lines 271: “metagenomics”

Response:

We apologize for our mistake and correct it in revised manuscript.

- In different figure 2, 4, 6 and 8 when you talk about coordination, muscle endurance and treadmill, information need to be more develop in the legend. We do not understand muscle endurance assessment through grip strength ? In the panel Treadmill should be replace by maximal running time.....

Response:

To make it clear, we changed ‘grip test’ into ‘grid hanging time’, ‘Coordination’ into ‘Maximal time on Rotarod’, ‘Treadmill’ into ‘Time to exhaustion’ in both panel and legend.

- Figure 3. Number of mice per group in panel c seems only to be 4 in low temperature group. If you excluded several mice? you need to bring justification.

Response:

We are sorry for the different number of samples, and we only sent 4 samples from LT mice to conduct 16S sequencing, since LT group was one of the control group. In published data, 31-day cold expose resulted in an increase of *Fimicutes* and a decrease of *Bacteroidetes*¹⁵. Similar changes were also showed in LT group, compared to RT group, indicating low temperature treatment exerted relatively stable effects on microbiota composition in mouse model.

Chevalier et al., 2015, Cell 163, 1360–1374

References:

- 1 J. J. Radley and P. E. Sawchenko, 'Evidence for involvement of a limbic paraventricular hypothalamic inhibitory network in hypothalamic-pituitary-adrenal axis adaptations to repeated stress', *JOURNAL OF COMPARATIVE NEUROLOGY*, 523: 18, 2015-12-15 2015, 2769-2787.
- 2 X. M. Ma, S. L. Lightman and G. Aguilera, 'Vasopressin and corticotropin-releasing hormone gene responses to novel stress in rats adapted to repeated restraint', *ENDOCRINOLOGY*, 140: 8, 1999-08-01 1999, 3623-3632.
- 3 Y. Deng and D. R. Paul, 'The Relationships Between Depressive Symptoms, Functional Health Status, Physical Activity, and the Availability of Recreational Facilities: a Rural-Urban Comparison in Middle-Aged and Older Chinese Adults', *INTERNATIONAL JOURNAL OF BEHAVIORAL MEDICINE*, 25: 3, 2018-06-01 2018, 322-330.
- 4 Y. Tian and Z. Shi, 'Effects of Physical Activity on Daily Physical Function in Chinese Middle-Aged and Older Adults: A Longitudinal Study from CHARLS', *Journal of Clinical Medicine*, 11: 21, 2022-11-02 2022.
- 5 K. D. Ono-Moore, I. M. Olfert, J. M. Rutkowski, S. V. Chintapalli, B. J. Willis, M. L. Blackburn, D. K. Williams, J. O'Reilly, T. Tolentino, KCK Lloyd and S. H. Adams, 'Metabolic physiology and skeletal muscle phenotypes in male and female myoglobin knockout mice', *Am J Physiol Endocrinol Metab*, 321: 1, 2021-07-01 2021, E63-E79.
- 6 K. D. Ono-Moore, J. M. Rutkowski, N. A. Pearson, D. K. Williams, J. L. Grobe, T. Tolentino, KCK Lloyd and S. H. Adams, 'Coupling of energy intake and energy expenditure across a temperature spectrum: impact of diet-induced obesity in mice', *Am J Physiol Endocrinol Metab*, 319: 3, 2020-09-01 2020, E472-E484.
- 7 T. Sato, Y. Ito and T. Nagasawa, 'L-Lysine suppresses myofibrillar protein degradation and autophagy in skeletal muscles of senescence-accelerated mouse prone 8', *BIOGERONTOLOGY*, 18: 1, 2017-02-01 2017, 85-95.
- 8 P. Palma-Granados, I. Seiquer, R. Benítez, C. Óvilo and R. Nieto, 'Effects of lysine deficiency on carcass composition and activity and gene expression of lipogenic enzymes in muscles and backfat adipose tissue of fatty and lean piglets', *Animal*, 13: 10, 2019-10-01 2019, 2406-2418.
- 9 C. L. Jin, J. L. Ye, J. Yang, C. Q. Gao, H. C. Yan, H. C. Li and X. Q. Wang, 'mTORC1 Mediates Lysine-Induced Satellite Cell Activation to Promote Skeletal Muscle Growth', *Cells*, 8: 12, 2019-11-30 2019.
- 10 J. Tokarz, G. Möller, A. Artati, S. Huber, A. Zeigerer, B. Blaauw, J. Adamski and K. A. Dyar, 'Common Muscle Metabolic Signatures Highlight Arginine and Lysine Metabolism as Potential Therapeutic Targets to Combat Unhealthy Aging', *INTERNATIONAL JOURNAL OF MOLECULAR SCIENCES*, 22: 15, 2021-07-26 2021.
- 11 G. Watanabe, H. Kobayashi, M. Shibata, M. Kubota, M. Kadowaki and S. Fujimura, 'Reduction in dietary lysine increases muscle free amino acids through changes in protein metabolism in chickens', *Poult Sci*, 99: 6, 2020-06-01 2020, 3102-3110.
- 12 U. S. Unni, T. Raj, S. Sambashivaiah, R. Kuriyan, S. Uthappa, M. Vaz, M. M. Regan and A. V. Kurpad, 'The effect of a controlled 8-week metabolic ward based lysine supplementation on muscle function, insulin sensitivity and leucine kinetics in young men', *CLINICAL NUTRITION*, 31: 6, 2012-12-01 2012, 903-910.
- 13 H. J. Lee, H. B. Jang, W. H. Kim, K. J. Park, K. Y. Kim, S. I. Park and H. J. Lee, '2-Aminoadipic acid (2-AAA) as a potential biomarker for insulin resistance in childhood obesity', *Sci Rep*, 9: 1,

2019-09-20 2019, 13610.

14 W. Y. Xu, Y. Shen, H. Zhu, J. Gao, C. Zhang, L. Tang, S. Y. Lu, C. L. Shen, H. X. Zhang, Z. Li, P. Meng, Y. H. Wan, J. Fei and Z. G. Wang, '2-Aminoadipic acid protects against obesity and diabetes', *JOURNAL OF ENDOCRINOLOGY*, 243: 2, 2019-11-01 2019, 111-123.

15 C. Chevalier, O. Stojanović, D. J. Colin, N. Suarez-Zamorano, V. Tarallo, C. Veyrat-Durebex, D. Rigo, S. Fabbiano, A. Stevanović, S. Hagemann, X. Montet, Y. Seimbille, N. Zamboni, S. Hapfelmeier and M. Trajkovski, 'Gut Microbiota Orchestrates Energy Homeostasis during Cold', *CELL*, 163: 6, 2015-12-03 2015, 1360-1374.

REVIEWERS' COMMENTS

Reviewer #1 (Remarks to the Author):

I congratulate the authors for their effort in addressing my concerns. I sincerely think that now the manuscript is greatly improved, and their interesting findings are much strongly supported by their results. I have only two minor points that I would like to mention to the authors.

First, when assessing autophagic flux *in vivo* (Fig S3d), the authors show a WB (and quantification) of autophagy markers in muscle extracts from mice untreated or treated with chloroquine (CQ). Usually, CQ treatment should lead to an increase in protein levels of LC3 and p62, due to the lack of degradation of these proteins through autophagy. Thus, in control mice + CQ, an increase in these proteins would be expected, which is not seen in the figure (if something happens is rather the opposite). Could the authors explain that?

Second, I suggest the authors to tone down their conclusion about inhibition of mitophagy (at least *in vivo*) in FT, FMTft and alpha-AA treated mice, since this parameter hasn't been evaluated directly, and it is based on the static expression of Park2, which is just one mitophagic protein. Again, flux experiments assessing different autophagic proteins in mitochondrial fractions would have been preferred to show that (i.e., LC3, BNIP3, Parkin, Pink1, among others). I appreciate the *in vitro* experiments, which clearly show an inhibition of mitophagy, but this is not sufficient to strongly state that *in vivo* there is an inhibition of mitophagy.

Reviewer #2 (Remarks to the Author):

Reviewer Comments:

General Comments:

The reviewer thinks that this original article entitled "Augmented temperature fluctuation aggravates muscular atrophy through the gut microbiota" is of interest. The authors correctly answer to reviewer comments and integrated it in the revision version.

For that reason I agree to accept the publication of the manuscript.

REVIEWERS' COMMENTS

Reviewer #1 (Remarks to the Author):

I congratulate the authors for their effort in addressing my concerns. I sincerely think that now the manuscript is greatly improved, and their interesting findings are much strongly supported by their results. I have only two minor points that I would like to mention to the authors.

First, when assessing autophagic flux in vivo (Fig S3d), the authors show a WB (and quantification) of autophagy markers in muscle extracts from mice untreated or treated with chloroquine (CQ). Usually, CQ treatment should lead to an increase in protein levels of LC3 and p62, due to the lack of degradation of these proteins through autophagy. Thus, in control mice + CQ, an increase in these proteins would be expected, which is not seen in the figure (if something happens is rather the opposite). Could the authors explain that?

Second, I suggest the authors to tone down their conclusion about inhibition of mitophagy (at least in vivo) in FT, FMTft and alpha-AA treated mice, since this parameter hasn't been evaluated directly, and it is based on the static expression of Park2, which is just one mitophagic protein. Again, flux experiments assessing different autophagic proteins in mitochondrial fractions would have been preferred to show that (i.e., LC3, BNIP3, Parkin, Pink1, among others). I appreciate the in vitro experiments, which clearly show an inhibition of mitophagy, but this is not sufficient to strongly state that in vivo there is an inhibition of mitophagy.

Response:

1) We appreciate Reviewer #1's the constructive comments. CQ administration often led to accumulation of LC3, as revealed in mitophagy flux of Figure 7c. In Fig S3d, there were no significant increase in P62 and LC3 after CQ administration might resulted from insufficient time of CQ administration (3 days), which led to inadequate inhibition of autophagy.

2) We agreed with Reviewer 1 that current experiments can't fully supporting the inhibited mitophagy level in vivo. Thus, we adjusted relevant statement in corresponding part (highlighted in yellow), to tone down the conclusion of inhibited mitophagy in animal models.

Reviewer #2 (Remarks to the Author):

Reviewer Comments:

General Comments:

The reviewer thinks that this original article entitled "Augmented temperature fluctuation aggravates muscular atrophy through the gut microbiota" is of interest.

The authors correctly answer to reviewer comments and integrated it in the revision version.

For that reason I agree to accept the publication of the manuscript.

Response:

We thank the reviewer for the constructive comments and kind evaluation.